# Weakly Supervised Video Scene Graph Generation via Natural Language Supervision

**Kibum Kim**[1]     **Kanghoon Yoon**[1]     **Yeonjun In**[1]     **Jaehyeong Jeon**[1]     **Jinyoung Moon**[2]
**Donghyun Kim**[3] **Chanyoung Park**[1]*
[1]KAIST     [2]ETRI     [3]Korea University
{kb.kim,ykhoon08,yeonjun.in,wogud405,cy.park}@kaist.ac.kr
jymoon@etri.re.kr   d_kim@korea.ac.kr

## Abstract

Existing Video Scene Graph Generation (VidSGG) studies are trained in a fully supervised manner, which requires all frames in a video to be annotated, thereby incurring high annotation cost compared to Image Scene Graph Generation (ImgSGG). Although the annotation cost of VidSGG can be alleviated by adopting a weakly supervised approach commonly used for ImgSGG (WS-ImgSGG) that uses image captions, there are two key reasons that hinder such a naive adoption: 1) *Temporality within video captions*, i.e., unlike image captions, video captions include temporal markers (e.g., before, while, then, after) that indicate time-related details, and 2) *Variability in action duration*, i.e., unlike human actions in image captions, human actions in video captions unfold over varying duration. To address these issues, we propose a **N**atural **L**anguage-based **V**ideo **S**cene **G**raph **G**eneration (NL-VSGG) framework that only utilizes the readily available video captions for training a VidSGG model. NL-VSGG consists of two key modules: Temporality-aware Caption Segmentation (TCS) module and Action Duration Variability-aware caption-frame alignment (ADV) module. Specifically, TCS segments the video captions into multiple sentences in a temporal order based on a Large Language Model (LLM), and ADV aligns each segmented sentence with appropriate frames considering the variability in action duration. Our approach leads to a significant enhancement in performance compared to simply applying the WS-ImgSGG pipeline to VidSGG on the Action Genome dataset. As a further benefit of utilizing the video captions as weak supervision, we show that the VidSGG model trained by NL-VSGG is able to predict a broader range of action classes that are not included in the training data, which makes our framework practical in reality. Our code is available at https://github.com/rlqja1107/NL-VSGG.

## 1 Introduction

Scene graph is a visually-grounded structured graph in which objects are represented as nodes and the relationships between them as directed edges. A scene graph bridges computer vision and natural language with high-level information, facilitating its usage on various downstream tasks, such as question answering (Ghosh et al., 2019), captioning (Yang et al., 2019), and retrieval (Schroeder & Tripathi, 2020).

In general, studies for scene graph generation (SGG) (Kim et al., 2024a; Yoon et al., 2023; Jeon et al., 2024; Yoon et al., 2024) have been conducted in the realm of images, referred to as **ImgSGG**. These studies primarily excel at predicting static relationships (e.g., standing on) within a single image, while struggling to predict dynamic relationships (e.g., running) that may exist over consecutive images, since ImgSGG models are unable to capture dynamic visual relations (Chen et al., 2023). In this regard, Video scene graph generation (Cong et al., 2021; Feng et al., 2023; Teng et al., 2021), dubbed as **VidSGG**, has emerged to capture temporal context across video frames and predict dynamic relationships, extending its scope beyond merely predicting static relationships within a single image.

Existing VidSGG studies (Cong et al., 2021; Nag et al., 2023; Xu et al., 2022; Wang et al., 2022a) follow a fully supervised approach, indicating a heavy reliance on costly annotation involving class

---

*Corresponding Author

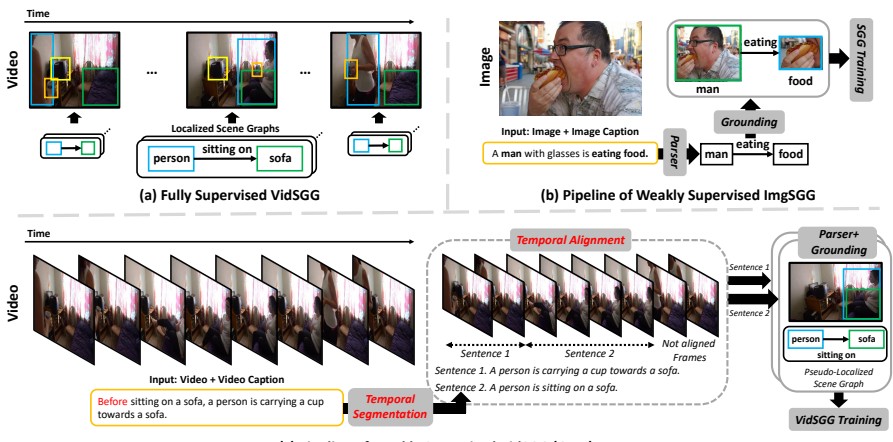

Figure 1: (a) The fully supervised VidSGG requires costly localized scene graphs across all frames. (b) The pipeline of WS-ImgSGG. (c) The pipeline of WS-VidSGG needs to consider the *temporality within the caption* addressed by temporal segmentation and the *variability of action duration* addressed by temporal alignment.

information (i.e., entity and relation) alongside bounding boxes for entities *across all frames* in a video (See Fig. 1(a)). This indicates that VidSGG requires greater annotation costs compared to ImgSGG, which only requires annotations for a single image. Despite recent weakly supervised ImgSGG (i.e., **WS-ImgSGG**) approaches (Zhong et al., 2021; Zhang et al., 2023; Ye & Kovashka, 2021; Kim et al., 2024b; Li et al., 2022a) that address the annotation cost in ImgSGG, weakly supervised approach for VidSGG (i.e., **WS-VidSGG**), where annotation costs are more demanding, remains unexplored. Although PLA (Chen et al., 2023), the first WS-VidSGG study, proposes a framework for training an VidSGG model based on a ground-truth unlocalized scene graph of the middle frame, we argue that the assumption of a ground-truth scene graph existing in the middle frame, among other frames, is not only unrealistic but also still requires manual human annotation, thus continuing to impose substantial annotation costs.

In this work, we are interested in training an VidSGG model without any human-annotated scene graphs, and there can be two different strategies. The first strategy would be to apply a pre-trained ImgSGG model, preferably the one with a strong zero-shot predictive ability for relations (e.g., RLIP (Yuan et al., 2022) and RLIPv2 (Yuan et al., 2023)), to every frame of the video to obtain pseudo-labeled scene graphs in each frame. However, since such a model is trained based on static visual relationships in images, they are not capable of predicting dynamic visual relationships[1]. The second strategy would be to leverage language supervision from video captions and follow the conventional WS-ImgSGG pipeline (Zhong et al., 2021; Zhang et al., 2023; Ye & Kovashka, 2021; Kim et al., 2024b; Li et al., 2022a) as depicted in Fig. 1(b). Specifically, we could first parse a video caption to extract triplets, align them with consecutive frames, and ground the aligned triplets within each frame. However, such a simple approach of adopting the WS-ImgSGG pipeline to VidSGG is limited[2] due to the following two key reasons:

- **Temporality within Video Captions.** Contrary to image captions shown in Fig. 1(b), video captions often contain temporal markers (e.g., before, while, then, after) representing time-related details. Without considering them, applying the above simple approach may erroneously supervise the model. For example, in Fig. 1(c), if we overlook the temporal marker before in the caption, and use <person, sitting on, sofa> to annotate earlier frames rather than later frames, the model would be mistakenly supervised, resulting in the performance degradation. As temporal markers account for around 65% of the Action Genome (AG) dataset used for VidSGG, while they account for only 2% in the Visual Genome (VG) (Krishna et al., 2017b) dataset used for ImgSGG (See Fig. 2), considering temporality is especially crucial in video captions.

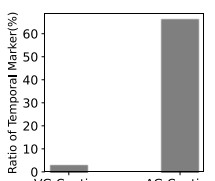

Figure 2: Ratio of temporal markers.

---

[1]In Table 1, we show that the performance of RLIP and RLIPv2 is subpar in the Action Genome (Ji et al., 2020) dataset under the zero-shot setting.

[2]In Table 1, we show that the simple approach performs poorly on the Action Genome dataset.

- **Variability in Action Duration.** Human actions unfold over varying duration in a video. For example, in Fig. 1(c), *Sentence 1* occurs within two frames, while *Sentence 2* extends across a longer span of four frames. However, if we overlook such variability in the action duration and naively align the first segmented sentence with the first four consecutive frames and the second sentence with the latter four consecutive frames (i.e., 8 frames ÷ 2 sentences), the 3rd and 4th frames would end up being annotated with <person, carrying, cup>, while the last two frames would end up being annotated with <person, sitting on, sofa>, both of which are undesired.

To this end, we propose a simple yet effective weakly-supervised VidSGG framework, called **N**atural **L**anguage-based **V**ideo **S**cene **G**raph **G**eneration (NL-VSGG), that only utilizes the readily available video captions for training a VidSGG model. Our proposed framework consists of two key modules: Temporality-aware Caption Segmentation (TCS) module, and Action Duration Variability-aware Caption-Frame Alignment (ADV) module. More precisely, TCS segments the video caption into multiple sentences while also considering the temporality within the video caption based on a Large Language Model (LLM). Then, ADV aligns the temporally segmented sentences with corresponding consecutive frames in the video considering the variability in action duration. The main idea is to perform $K$-means clustering on the frames, and assign each segmented sentence to the frames within a cluster based on its similarity with the cluster centroids.

Through our extensive experiments, we demonstrate the superiority of NL-VSGG over the simple adoption of 1) a pre-trained ImgSGG model, and 2) the WS-ImgSGG pipeline to VidSGG. It is worth noting that for the first time, we show the capability of training a VidSGG by only utilizing the readily available video captions, i.e., language supervision. As a further benefit of utilizing the video captions as weak supervision, we show that the VidSGG model trained by NL-VSGG is able to predict a broader range of action classes that are not included in the training data, which makes our framework practical in reality. Our contributions are summarized as follows:

- We identify two key reasons for why a simple adoption of the WS-ImgSGG pipeline to VidSGG fails, i.e., temporality within video captions and variability of action duration.

- We propose a simple yet effective weakly supervised VidSGG framework, called NL-VSGG, that only utilizes the readily available video captions for training. To the best of our knowledge, we are the first to enable the training of VidSGG model with language supervision without manual annotation.

- Our extensive experiments on the Action Genome dataset demonstrate the superiority of NL-VSGG. Our proposed method is practical in that utilizing the video captions as weak supervision allows the VidSGG model to be able to predict action classes that are not included in the training data.

## 2 WS-VIDSGG WITH NATURAL LANGUAGE SUPERVISION

In this section, we describe the pipeline of NL-VSGG for training a VidSGG model based on natural language supervision of video captions. We start by outlining the problem formulation of WS-VidSGG (Section 2.1). Subsequently, we describe Temporality-aware Caption Segmentation (TCS) module that segments a video caption in a temporal order via an LLM (Section 2.2), followed by Action Duration Variability-aware Caption-Frame Alignment (ADV) module that aligns each of the segmented sentence with appropriate frames (Section 2.3). Then, we parse the aligned segmented sentences to extract triplets, ground them within each frame, and train the VidSGG model with pseudo-localized scene graphs (Section 2.4). Finally, we describe a novel pseudo-labeling strategy that leverages negative action classes using motion cues within unaligned frames (Section 2.5). The overall framework is shown in Fig. 3.

### 2.1 PROBLEM FORMULATION

In this work, given a video $V = \{I^1, I^2, ..., I^T\}$ and its paired video caption $S$, where $I^t$ is the $t$-th frame of the video $V$ and $T$ is the number of video frames in the video $V$, our goal is to generate a scene graph $G^t = \{s_j^t, p_j^t, o_j^t\}_{j=1}^{N^t}$ for each frame $I^t$, where $N^t$ is the number of triplets in $I^t$. Moreover, $s_j^t/o_j^t$ denote the subject/object, whose entity classes are $s_{j,c}^t/o_{j,c}^t \in \mathcal{C}^e$, and their bounding boxes are given by $s_{j,b}^t/o_{j,b}^t$. $p_j^t$ denotes the action of $s_j^{t3}$ interacting with $o_j^t$, and its class is denoted by $p_{j,c}^t \in \mathcal{C}^a$, where $\mathcal{C}^e$ and $\mathcal{C}^a$ are predefined entity classes and action classes,

---

[3]In the Action Genome dataset, the subject is always a person.

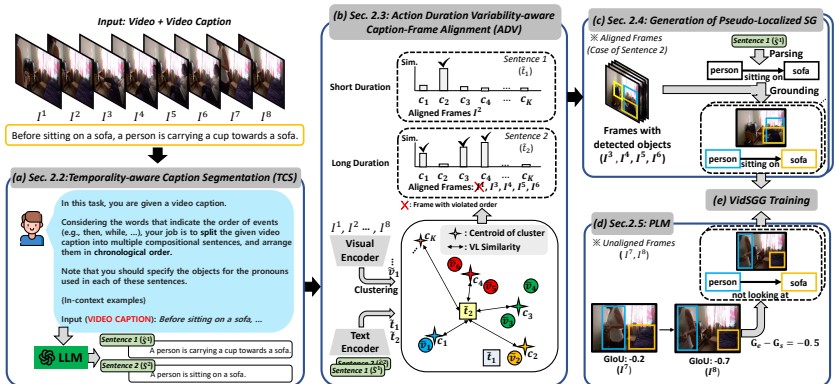

Figure 3: The overall framework of NL-VSGG. With an input video and its caption, (a) we employ the TCS module to segment the input video caption into sentences based on temporality. (b) In the ADV module, each segmented sentence is aligned with appropriate frames considering the variability in action duration. (c) The segmented sentences are then parsed and grounded to generate pseudo-localized scene graphs. (d) Furthermore, we assign negative classes based on the motion cues within unaligned frames. (e) Utilizing the pseudo-localized scene graphs and pseudo-labeled negative classes, we then train a VidSGG model.

respectively. Given a scene graph $G^t$ across all frames along the time axis (i.e., $t \in \{1, 2, ...T\}$), the scene graph for video $V$ is represented by $\mathcal{G} = \{G^1, G^2, ..., G^T\}$.

**Difference with existing fully/weakly supervised approaches.** Note that a fully supervised approach (Cong et al., 2021; Nag et al., 2023) relies on a localized video scene graph $\mathcal{G}$ for training the model. On the other hand, a recently proposed weakly supervised approach (i.e., PLA (Chen et al., 2023)) relies on a ground-truth unlocalized scene graph of the middle frame, i.e., $G^{t'} = \{s_j^{t'}, p_j^{t'}, o_j^{t'}\}_{j=1}^{N^{t'}}$, where $t'$ is $T/2$ and the bounding boxes $s_{j,b}^{t'}/o_{j,b}^{t'}$ are not provided. Both fully/weakly supervised approaches require structured scene graphs that demand costly human labor, while NL-VSGG only necessitates a readily available video caption $S$.

## 2.2 TEMPORALITY-AWARE CAPTION SEGMENTATION (TCS)

In this module, we segment the video caption $S$ considering the temporal order of events to clearly understand the sequence of actions. To this end, we employ an LLM based on a prompt that is designed to be aware of temporality within video captions. Our newly designed prompt is described as follows. First, the initial instruction to inform the LLM about the task at hand while enforcing it to consider the temporality of events occuring in the video caption is designed as: *Your job is to split the given video caption into multiple compositional sentences and arrange them in chronological order.* Moreover, video captions often require coreference resolution (Peng et al., 2019) caused by a pronoun referring to the same object over time, which may hinder an accurate extraction of triplets. To this end, we provide an additional prompt: *Note that you should specify the objects for the pronouns used in each of these sentences.* Following the above prompts, we provide a few examples related to the task at hand (i.e., in-context few-shot learning (Brown et al., 2020)) to further engage the LLM. Finally, we instruct the LLM to segment the video caption $S$ in a temporal order, which in turn yields segmented sentences $\{\hat{S}^1, \hat{S}^2, .., \hat{S}^m, ..., \hat{S}^M\}$, where $M$ is the number of segmented sentences and $M < T$. For example, in Fig. 3, *Sentence 1* and *Sentence 2* correspond to $\hat{S}^1$ and $\hat{S}^2$, respectively, with $M$ being 2. In summary, we design prompts tailored to the WS-VidSGG task particularly focusing on capturing the temporality within the video caption and addressing the coreference issue. Please refer to Appendix A regarding the details of the prompt, and Appendix B regarding the impact of the prompt addressing the coreference resolution.

## 2.3 ACTION DURATION VARIABILITY-AWARE CAPTION-FRAME ALIGNMENT (ADV)

Having obtained the segmented sentences $\{\hat{S}^1, \hat{S}^2, .., \hat{S}^m, ..., \hat{S}^M\}$ from the video caption $S$, we need to align each segmented caption $\hat{S}^m$ with frames that visually correspond to the scene being described therein. However, this alignment process requires careful attention to ensure effective supervision of the model, taking into account the variability in action duration, as discussed in Section 1. In doing so, it is crucial to estimate how the visual semantic of each frame $I^1, I^2, ..., I^T$

is relevant to the textual semantic of each segmented sentence $\hat{S}^m$. Hence, we employ a vision-language model[4] that captures the joint space of visual and textual semantics. More precisely, we feed $I^1, I^2, ..., I^T$ into a visual encoder $f_{vis}$ to extract visual features (i.e., $\tilde{v}_t = f_{vis}(I^t)$) and $\hat{S}^m$ into a text encoder $f_{text}$ to extract textual features (i.e., $\tilde{t}_m = f_{text}(\hat{S}^m)$), followed by $K$-Means clustering algorithm on $\{\tilde{v}_1, \tilde{v}_2, ..., \tilde{v}_T\}$ to generate $K$ proposals with which the textual features can be aligned. The most straightforward approach to align $\tilde{t}_m$ with corresponding video frames would be to compute the similarity scores between $\tilde{t}_m$ and the centroids $\{c_1, c_2, ..., c_K\}$, and select the frames near the most similar centroid. However, this approach cannot effectively capture the variability of action duration, especially in long action duration cases, e.g., *Sentence 2* in Fig. 3. To this end, we propose a simple yet effective method of leveraging the variation in similarity scores.

**Clustering-based Caption-Frame Alignment.** For a sentence with long action duration, its corresponding frames would span across multiple clusters, exhibiting relatively high similarity scores with multiple cluster centroids. On the other hand, a sentence with short action duration would represent a confidently sharpened similarity score mainly focusing on a single cluster. Hence, for the representation $\hat{t}^m$ of each segmented sentence $\hat{S}^m$, we arrange its similarity scores with $K$ centroids in descending order, and determine the point where the scores show the steepest decline. Note that this steepest decline point will determine the varying durations of actions by isolating the relevant multiple clusters, regardless of how long the action lasts. Then, we choose all the clusters preceding this point, and align the frames in those clusters with $\hat{t}^m$. For example, for *Sentence 2* in Fig. 3, we sort the cluster based on the similarity scores (i.e., $c_4, c_3, c_1, c_2,...$) and determine the point where the score shows a steepest decline (i.e., $c_1 \rightarrow c_2$). Then, we choose frames in clusters $c_4, c_3$, and $c_1$ to align with *Sentence 2*. In this manner, we can adaptively choose clusters according to the variability of action duration. Regarding the $K$, our aim is to adapt the number of clusters depending on the length of the video. Thus, we set $K$ as $\frac{|V|}{\beta}$, where $\beta$ is a hyperparameter.

**Removing Unrealistic Caption-Frame Alignments.** It is worth noting that we remove unrealistic alignments that violates the temporal order. For example, if $\hat{S}^1$ is aligned with $I^2$, and $\hat{S}^2$ is aligned with $I^1, I^3, I^4, I^5, I^6$, we remove $I^1$ from $\hat{S}^2$ since $I^2$ is already aligned with $\hat{S}^1$, which precedes $\hat{S}^2$ (See Fig. 3). Finally, each segmented sentence $\hat{S}^m$ is aligned with a sequence of consecutive frames within $V = \{I^1, I^2, ..., I^T\}$, i.e., $[I]^{\hat{S}^m}$.

## 2.4 GENERATION OF PSEUDO-LOCALIZED SCENE GRAPHS

Based on the $\{(\hat{S}^m, [I]^{\hat{S}^m})\}_{m=1}^M$ obtained from ADV, we construct pseudo-localized scene graphs for training the model, which can be described as follows:

**Scene graph parsing.** We begin by converting each segmented sentence $\hat{S}^m$ into a triplet $<s_m, p_m, o_m>$[5] using either a rule-based parser (Schuster et al., 2015) or an LLM-based parser (Kim et al., 2024b). Note that the bounding boxes $s_{m,b}, o_{m,b}$ are not defined, and the classes $s_{m,c}, o_{m,c}, p_{m,c}$ are not necessarily included in $\mathcal{C}^e$ and $\mathcal{C}^a$. To ensure $s_{m,c}, o_{m,c} \in \mathcal{C}^e$ and $p_{m,c} \in \mathcal{C}^a$, we map them to the classes of our interest (i.e., entity/action classes in the Action Genome dataset) by either using synsets' lemmas and hypernyms in WordNet (Miller, 1995) or LLM-based alignment (Kim et al., 2024b). In this process, we obtain pseudo-unlocalized scene graphs. Note that we can also omit the class mapping process to let the model be able to predict a broader range of action classes that are not included in the entity/action classes of Action Genome, as will be shown in Section 3.6.

**Scene graph grounding.** To define the bounding boxes $s_{m,b}$ and $o_{m,b}$, we follow a prior study (Chen et al., 2023) that leverages the information of a pretrained object detector. Specifically, we ground $s_m$ to a detected bounding box whose entity class corresponds to $s_{m,c}$, while $o_m$ is grounded in a similar manner. After grounding $s_m$ and $o_m$, we assign $p_m$ between $s_m$ and $o_m$.

The above processes of scene graph parsing and scene graph grounding are adopted to each pair in $\{(\hat{S}^m, [I]^{\hat{S}^m})\}_{m=1}^M$, where the scene graph grounding process is applied across aligned frames $[I]^{\hat{S}^m}$, after which pseudo-localized video scene graphs $\mathcal{G}$ can be obtained.

---

[4]For the vision-language model, we used DAC (Doveh et al., 2024), a variant of CLIP Radford et al. (2021), due to its compositional reasoning ability. Moreover, the vision-language model can be replaced with a video-language model (Refer to Appendix D for experiments that replaces DAC with InternVideo).

[5]Although each segmented caption can be converted into more than one triplet, we assume here that it is converted to only one triplet for simplicity of explanation.

## 2.5 OPTIONAL: DEALING WITH NEGATIVE ACTION CLASSES IN ACTION GENOME

It is important to note that while the Action Genome dataset contains negative action classes (e.g., not looking at and not contacting), video captions usually do not contain negations[6]. For this reason, a VidSGG model trained in a weakly supervised manner based on video captions as in our study would fail to generate a scene graph containing negative action classes in the Action Genome dataset. To address this issue, we propose a novel

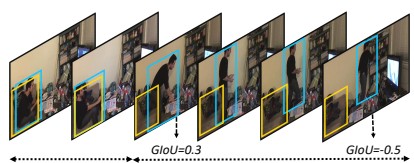

Figure 4: Example of motion cue.

Pseudo-Labeling strategy based on Motion cues (PLM) that assigns negative action classes between subject-object pairs. The core idea is to make use of the fact that a person usually does not look at (or contact) an object when moving apart from it. Specifically, we adopt the generalized IoU (Rezatofighi et al., 2019; Gao et al., 2023) (GIoU) as a metric to measure how close two objects are from each other. Note that for a given subject and an object, a small GIoU in the end frame (i.e., $\mathbf{G}_e$) and a large GIoU in the start frame (i.e., $\mathbf{G}_s$) indicates that they are getting farther from each other over time. For example, as a person is moving farther from a sofa from the 3rd frame to the last frame, the GIoU gets smaller (See Figure 4). In other words, as $\mathbf{G}_e - \mathbf{G}_s$ gets smaller, the subject and the object are getting farther from each other over time. With this motion cue, we assign negative action classes, i.e., not looking at and not contacting.

However, as considering all the frames in a video for pseudo-labeling of negative action classes is costly, we utilize frames that are not aligned with any of the segmented sentences for pseudo-labeling negative action classes. More precisely, for all the unaligned frames across the entire videos given in the dataset, we compute $\mathbf{G}_e - \mathbf{G}_s$ between the same subject-object pair. Then, we sort the computed GIoUs in the ascending order, and assign pseudo-labels (i.e., not looking at and not contacting) to top-$\alpha\%$ aiming at assigning labels to more confident subject-object pairs. More precisely, we restrict the assignment of not looking at to the start and end frames within unaligned frames, and not contacting only to the end frames. Please refer to Appendix E regarding the different selection strategies for pseudo-label assignment. Since considering all subject-object pairs included in the training data is not only time-consuming but also prone to noise, we only select objects appearing in the pseudo-localized video scene graph $\mathcal{G}$ obtained from the aligned frames in each video.

It is important to note that the proposed PLM is an optional module that can be only applied when the dataset contains negative classes as in the Action Genome dataset. While we show its effectiveness on the original Action Genome dataset with negative classes, we also provide results for datasets without negative classes, namely the AG dataset without negative classes See Table 10 in Appendix I) and VidHOI dataset (Chiou et al., 2021) (See Table 6 in Appendix C).

**Model Training.** Lastly, having obtained the pseudo-localized video scene graph $\mathcal{G}$ and pseudo-labeled negative classes within unaligned frames, we then follow the training protocol of existing VidSGG models, i.e., STTran (Cong et al., 2021) and DSG-DETR (Feng et al., 2023).

## 3 EXPERIMENT

### 3.1 EXPERIMENTAL SETTING

**Datasets.** To **train** a VidSGG model without ground-truth localized video scene graphs[7], we use three video caption datasets: the Action Genome (Ji et al., 2020) (AG) caption, the MSVD (Chen & Dolan, 2011) caption, and the ActivityNet caption (Krishna et al., 2017a) datasets. For the AG caption dataset, we use 7,454 videos consisting of 166,785 frames, following previous studies (Chen et al., 2023; Cong et al., 2021; Ji et al., 2020). The duration of each video is on average 29.9 seconds. To show a more practical setting, which is to use external caption datasets, other than a benchmark VidSGG dataset containing video captions (i.e., AG dataset), as weak supervision, we use the MSVD and ActivityNet caption datasets, each of which contains 1,970 videos consisting of 40,863 frames with 2 FPS and 4,640 videos consisting of 569,836 frames with 1 FPS, respectively. The duration of each video for the MSVD and ActivityNet datasets is on average 9.5 and 117.3 seconds, respectively. Note that we mainly use the AG caption dataset for analysis throughout this paper while the MSVD and ActivityNet caption datasets are only used in Section 3.4 and 3.5. To **evaluate** our proposed NL-VSGG framework, we use the AG dataset containing ground-truth localized video scene graphs with 36 object classes (i.e., $\mathcal{C}^e$) and 25 action classes (i.e., $\mathcal{C}^a$), whose categories are divided into

---

[6]Only 0.09% (11/11,593) among all captions in Action Genome contain negation.

[7]The ground-truth localized video scene graphs are utilized for model training in a fully-supervised approach while we do not use them for weakly supervised approach.

three types, i.e., 3 attention classes, 6 spatial classes, and 16 contacting classes. Following previous studies (Chen et al., 2023; Cong et al., 2021), we use 1,747 videos with 54,429 frames. Furthermore, to validate the generalization on other datasets, we train and evaluate our proposed framework using the VidHOI (Chiou et al., 2021) dataset, which is detailed in Appendix C.

**Evaluation Metrics.** We use the widely adopted Recall@K (R@K) with "With Constraint" and "No Constraint" strategies, following previous studies (Cong et al., 2021; Chen et al., 2023; Feng et al., 2023). "With Constraint" allows only one action prediction with the highest score between subject and object, while "No Constraint" reflects the multi-label prediction capability in evaluation metric as the AG dataset follows a multi-label task. For example, if multiple high action scores are associated with a single subject-object pair, all of them are allowed in the top-K. Regarding the evaluation task, we adopt the Scene Graph Generation (SGDet) task, which is commonly used for WS-ImgSGG (Kim et al., 2024b; Zhong et al., 2021; Zhang et al., 2023; Ye & Kovashka, 2021). In this task, the ground-truth (GT) bounding boxes and entity class information are not provided during evaluation, and the predicted triplet is considered valid only if the predicted bounding box overlaps with the GT bounding box with an IoU>0.5.

**Baselines.** To evaluate our proposed framework, we compare it with models each of which corresponds to one of the four types of supervision: **Zero-shot** supervision[8] (No supervision), **Full** supervision, **Weak** supervision of **GT Unlocalized scene graph** (SG), and **Weaker** supervision of **Natural Language**, i.e., utilization of readily available video captions. Specifically, zero-shot supervision indicates that without any training process, each frame is considered as a static image and inferred with ImgSGG models equipped with strong zero-shot predictive ability for relations. For this purpose, we employ RLIP (Yuan et al., 2022) and RLIPv2 (Yuan et al., 2023). Full supervision, which we consider as the upper bound of our model, involves training the model with ground-truth localized video scene graphs, and we leverage STTran (Cong et al., 2021) and DSG-DETR (Feng et al., 2023) for VidSGG models. Weak supervision of GT unlocalized SG is to utilize a GT unlocalized scene graph in the middle frame, which is proposed by PLA (Chen et al., 2023). Moreover, we also compare with $PLA_{simp.}$, a simplified version of PLA trained using only unlocalized SG from the middle frame without any component of PLA. This baseline is included to demonstrate the heavy reliance of PLA on the annotated GT unlocalized scene graph, which is costly. Finally, the natural language supervision is based on the readily available video captions. It includes two approaches: straightforward approach of the WS-ImgSGG method discussed in Section 1 (i.e., + WS-ImgSGG) and our proposed framework (i.e., + NL-VSGG). In addition to those approaches, we add a new baseline adapted from PLA (i.e, + $PLA_{cap}$), where the GT unlocalized scene graph of the middle frame are replaced with pseudo-unlocalized scene graphs obtained from video captions. It is important to note that video captions provides a *weaker* supervision signal to the VidSGG model compared with the GT unlocalized SG (hence we name it "Weaker").

**Implementation Details.** For the pretrained object detector, we follow PLA (Chen et al., 2023), which employs VinVL (Zhang et al., 2021) with backbone ResNeXt-152 C4. This object detector only leaves bounding boxes for objects with a confidence score of 0.2 or higher. In the TCS module, we use *gpt-3.5-turbo* in ChatGPT (OpenAI, 2023) for an LLM. In the ADV module, DAC (Doveh et al., 2024) is adopted for a vision-language model. Please refer to Appendix D regarding the experiment with an open-source smaller language model (Jiang et al., 2023) and another vision-language model (Wang et al., 2022c). Additionally, $\beta$ used to determine the number of clusters $K$ is set to 4, and $\alpha$ used in the pseudo-labeling strategy is set to 15%. Please refer to Appendix J regarding the sensitivity of hyperparameter $\beta$ and $\alpha$. Regarding the triplet extraction in Section 2.4, we adopt an LLM-based approach (Kim et al., 2024b). Please refer to Appendix F regarding the result of different triplet extraction processes. For the experimental environment, we implemented NL-VSGG on both an NVIDIA GeForce A6000 48GB and an Intel Gaudi-v2.

### 3.2 QUANTITATIVE RESULTS

Table 1 shows the performance comparisons across four types of supervision, utilizing STTran (Cong et al., 2021) and DSG-DETR (Feng et al., 2023) as backbones for full and weak/weaker supervision. We have the following observations: **1)** Models with zero-shot predictive ability for relation, such as RLIP (Yuan et al., 2022) and RLIPv2 (Yuan et al., 2023), show inferior performance, especially in With Constraint setting. This suggests that they struggle to predict key relationships between subject-object pairs, highlighting the limitation of simply applying ImgSGG models to the VidSGG task, as these models fail to account for dynamic relationships. Please refer to Appendix G for experiments regarding the incorporation of dynamic relationships into the RLIP.

---

[8]We use the term zero-shot setting interchangeably with zero-shot supervision.

**2)** $PLA_{simp.}$, which solely relies on a GT unlocalized SG of the middle frame without any components proposed in PLA, shows competitive performance with the vanilla PLA (Chen et al., 2023), while significantly surpassing the performance of RLIP/RLIPv2, which are trained on a large number of images with complex models. This suggests that

Table 1: Results of four types of supervision on the AG dataset.

| Backbone | Method | Supervision | With Constraint | | No Constraint | |
|---|---|---|---|---|---|---|
| | | | R@20 | R@50 | R@20 | R@50 |
| RLIP RLIPv2 | ImgSGG | Zero-shot | 7.93 8.37 | 9.16 10.05 | 9.70 14.60 | 13.80 21.42 |
| STTran | Vanilla | Full | 33.98 | 36.93 | 36.20 | 48.88 |
| | +PLA +PLA$_{simp.}$ | Weak (GT Unlocalized SG) | 20.94 20.42 | 25.79 25.43 | 22.34 21.72 | 31.69 30.87 |
| | +WS-ImgSGG +PLA$_{cap}$ +NL-VSGG | Weaker (Natural Language) | 10.01 10.40 **15.61** | 12.83 13.26 **19.60** | 9.02 10.64 **15.92** | 14.05 15.13 **22.56** |
| DSG-DETR | Vanilla | Full | 34.80 | 36.10 | 40.90 | 48.30 |
| | +PLA +PLA$_{simp.}$ | Weak (GT Unlocalized SG) | 21.30 20.78 | 25.90 25.79 | 22.70 22.31 | 31.90 31.69 |
| | +WS-ImgSGG +PLA$_{cap}$ +NL-VSGG | Weaker (Natural Language) | 10.05 10.36 **15.75** | 12.96 13.53 **20.40** | 10.29 10.57 **16.11** | 14.77 15.41 **23.21** |

incorporating even a few ground-truth unlocalized video scene graphs is essential for strong performance in the VidSGG task, while also revealing PLA's heavy reliance on annotated GT unlocalized scene graphs, which are expensive to obtain. This is further verified by the significant performance drop when the GT unlocalized scene graph of the middle frame is replaced with a pseudo-unlocalized scene graph obtained from the video caption (PLA vs. $PLA_{cap}$). Moreover, we argue the assumption of a GT scene graph existing in the middle frame among other frames is not only unrealistic but also still requires manual human annotation, which makes these methods impractical in reality. **3)** Given the natural language supervision from video captions, which further relieves the annotation cost of PLA, NL-VSGG exhibits notably superior performance compared with a simple adoption of the WS-ImgSGG pipeline to VidSGG and $PLA_{cap}$, both of which disregard the two key factors discussed in Section 1, i.e., temporality within video captions and variability in action duration.

### 3.3 ABLATION STUDIES

In Table 2, we conduct ablation studies on the AG dataset to verify the effectiveness of each module in NL-VSGG. For the ablation studies, we use STTran (Cong et al., 2021) as the backbone. The variant of not using

Table 2: Ablation studies on each module of NL-VSGG.

| Row | TCS | ADV | PLM | With Constraint | | No Constraint | | Mean | |
|---|---|---|---|---|---|---|---|---|---|
| | | | | R@20 | R@50 | R@20 | R@50 | R@20 | R@50 |
| (a) | | | | 10.01 | 12.83 | 9.02 | 14.05 | 9.52 | 13.44 |
| (b) | ✓ | | | 11.09 | 14.66 | 11.34 | 16.70 | 11.22 | 15.68 |
| (c) | ✓ | ✓ | | 11.98 | 15.58 | 11.93 | 17.36 | 11.96 | 16.47 |
| (d) | ✓ | ✓ | ✓ | **15.61** | **19.60** | **15.92** | **22.56** | **15.77** | **21.08** |

any module (row (a)) corresponds to a simple approach of adopting the WS-ImgSGG pipeline to the VidSGG task. We have the following observations: **1. Effect of TCS:** We observe that the adoption of the TCS module to capture the temporality within the video captions enhances the performance compared with a simple approach (row (a) vs. (b)). Note that we apply the TCS module to the simple approach by considering each triplet as a segmented sentence. The performance enhancement demonstrates the effectiveness of the TCS module in capturing the temporality, leading to accurate supervision for the model. **2. Effect of ADV:** We observe that the incorporation of the ADV module responsible for capturing the variability in action duration further enhances the performance (row (b) vs. row (c)), which demonstrates the importance of considering the variability of human actions. Regarding a qualitative analysis of the ADV module, please refer to Appendix H. **3. Effect of PLM:** Our proposed pseudo-labeling strategy with motion cues significantly improves the performance (row (c) vs. (d)), demonstrating the effectiveness of pseudo-labeling negative action classes. We recognize that the PLM module contributes the most to the the final performance, since the negative action classes, i.e., not looking at and not contacting, belong to head predicate classes, accounting for 16.5% and 8.7% of the predicates in the test set, respectively. Recall that PLM is an optional module that can be only applied when the dataset contains negative classes. Thus, to more precisely validate the effectiveness of the TCS and ADV modules, we intentionally removed negative classes from both training and test sets in the AG dataset, and show the results in Appendix I.

### 3.4 EXPLORING THE POTENTIAL OF UTILIZING EXTERNAL VIDEO-TEXT DATASET

Note that our proposed framework is not only limited to a benchmark dataset, i.e., Action Genome (Ji et al., 2020) (AG). In this section, we explore the potential of utilizing an external video-text paired dataset, i.e., MSVD (Chen & Dolan, 2011), for training. For this experiment, we use STTran as the backbone. In Table 3, we observe that

Table 3: Performance when an external video-text dataset is utilized for training.

| Training Dataset (Tested on the AG) | With Constraint | | No Constraint | |
|---|---|---|---|---|
| | R@20 | R@50 | R@20 | R@50 |
| AG | 15.61 | 19.60 | 15.92 | 22.56 |
| MSVD | 9.05 | 11.31 | 10.22 | 16.60 |
| AG+MSVD | **15.71** | **20.00** | **16.07** | **23.21** |

the performance based on sole utilization of the MSVD caption dataset for training shows inferior performance compared to that of AG caption.

This is expected since the video domains of these datasets are very different (i.e., indoor scenes in AG and outdoor scenes in MSVD). However, we observe that the combination of AG and MSVD caption datasets performs the best, implying that collecting a large pool of video-text pair regardless of the domains is helpful. In this regard, we believe this study paves the way of enhancing the inherent limited VidSGG datasets.

## 3.5 Analysis of performance on longer video dataset

To explore the impact of video length, we conducted experiments using the ActivityNet (Krishna et al., 2017a) caption dataset (average length: 117.3 seconds), which is approximately 4 times longer than the Action Genome caption dataset (average length: 29.9 seconds) and 12 times longer than the MSVD

Table 4: Performance over various video length. We use backbone as STTran.

| Training Dataset (Caption) | Method | Avg. Video Length | With Constraint | | No Constraint | | Mean |
|---|---|---|---|---|---|---|---|
| | | | R@20 | R@50 | R@20 | R@50 | |
| Action Genome | WS-ImgSGG | 29.9 seconds | 10.01 | 12.83 | 9.02 | 14.05 | 11.48 |
| | NL-VSGG | | 15.61 | 19.60 | 15.92 | 22.56 | 18.42 |
| MSVD | WS-ImgSGG | 9.5 seconds | 6.22 | 8.03 | 7.69 | 12.31 | 8.56 |
| | NL-VSGG | | 9.05 | 11.31 | 10.22 | 16.60 | 11.80 |
| ActivityNet | WS-ImgSGG | 117.3 seconds | 10.86 | 14.47 | 10.07 | 15.80 | 12.80 |
| | NL-VSGG | | 13.46 | 17.58 | 13.94 | 21.41 | 16.60 |

dataset (average length: 9.5 seconds). As shown in Table 4, we made the following two observations: 1) Regardless of video length, our proposed method consistently outperformed the naive approach (i.e., WS-ImgSGG). This indicates that our proposed method remains effective across videos of various lengths. 2) When comparing the performance between the MSVD and ActivityNet datasets, aside from the benchmark dataset (i.e., Action Genome), we observed that both WS-ImgSGG and NL-VSGG achieved better performance on the longer ActivityNet dataset compared to the shorter MSVD dataset. We attribute it to the fact that longer videos allow the model to learn more diverse video content, thereby improving generalization, and provide more supervision as the duration of actions increases. In this context, despite the shorter video length of the Action Genome dataset compared to the ActivityNet dataset, our proposed method performs better on the Action Genome dataset. This is because the video distribution in the Action Genome dataset is more closely aligned with the test set, which is derived from Action Genome.

## 3.6 Qualitative Results for Expansion of Action Classes

Recall that we can allow a VidSGG model trained by NL-VSGG to be able to predict a broader range of action classes that are not included in the training data (i.e., AG dataset), as our framework utilizes the video captions as weak supervision. To validate this, we perform qualitative analysis with STTran backbone on the AG test set. Fig. 5 shows qualitative results for scene graphs gen-

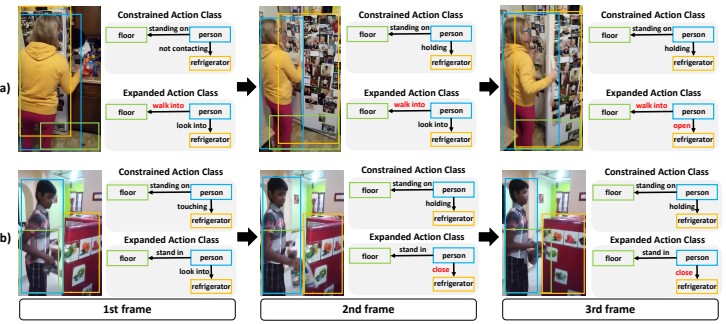

Figure 5: Qualitative results of NL-VSGG for broader range of action classes. The red-colored texts indicate predicates with novel meanings that are not present in the AG dataset.

erated under the following two conditions: an VidSGG model trained on 1) constrained action classes[9] that are obtained through the class mapping process described in Section 2.4, and 2) expanded action classes (i.e., 500 classes in total[10]) that are obtained without the class mapping process. In Fig. 5(a), we observe that the expansion of action classes enhances the temporal coherence of video scene graphs in that in the 3rd frame, the prediction of walking into/open under expanded action classes carries a stronger temporal implication compared to the prediction of standing on/holding under constrained classes. Furthermore, we observe that in Fig. 5(b), the expansion of action classes allows for predicting close, which presents the opposite meaning to open in Fig. 5(a), while the VidSGG model trained on the constrained classes still generate holding. This indicates that the expansion of action classes with temporal implication significantly benefits VidSGG models, which are capable of capturing temporal context, thereby generating video scene graphs with enhanced temporal coherence. In this vein, we believe that our proposed framework facilitates the training of VidSGG models with temporal coherence.

---

[9]Among the three types of action classes (i.e., attention, spatial and contacting), we generate scene graphs with the contacting class type, which conveys more descriptive information than the other two types.

[10]To filter out noisy classes, we select the top 500 most frequent action classes from 2,325 action classes. Refer to Appendix K regarding for more information on 500 action classes.

## 4 RELATED WORK

### 4.1 VIDEO SCENE GRAPH GENERATION (VIDSGG)

Video Scene Graph Generation (VidSGG) aims to learn the spatio-temporal dependencies of a video to predict the dynamic relationships between all object pairs. Existing VidSGG models are categorized into two settings based on the granularity of the generated video scene graph: video-level VidSGG (Shang et al., 2017; Tsai et al., 2019; Shang et al., 2021; Wei et al., 2024) and frame-level VidSGG (Cong et al., 2021; Feng et al., 2023; Teng et al., 2021; Li et al., 2022b). Video-level VidSGG models generate a global scene graph for a video clip, while frame-level VidSGG models generate a scene graph for each frame in a video clip. Note that our work follows the frame-level VidSGG setting. Following the release of the Action Genome dataset (Ji et al., 2020), frame-level VidSGG models have been actively researched. Specifically, STTran (Cong et al., 2021) employs two separate transformers to capture spatio-temporal dependencies of objects and frames, and TRACE (Teng et al., 2021) proposes a hierarchical relation tree method to enhance spatial-temporal reasoning. DSG-DETR (Feng et al., 2023) mitigates the complexity issue arising from the fully-connected graph between frames, enabling the capture of long-term temporal context. Recently, TEMPURA (Nag et al., 2023) and FloCoDe (Khandelwal, 2024) aim to alleviate the long-tailed predicate issue through memory-guided learning and label correlation loss, respectively. However, as these methods heavily rely on costly annotations on all frames, we propose a weakly supervised approach for VidSGG, which utilizes the readily available video captions, i.e., language supervision, for the first time.

### 4.2 WEAKLY SUPERVISED SCENE GRAPH GENERATION

**Weakly Supervised ImgSGG (WS-ImgSGG).** To relax a heavy reliance on the costly annotation of a fully supervised approach, ImgSGG utilizes two types of weak supervision. **1) Unlocalized scene graph**: It uses ground-truth scene graphs represented in text format, which have not yet been grounded to bounding boxes. In this regard, related studies (Ye & Kovashka, 2021; Shi et al., 2021; Zareian et al., 2020; Li et al., 2022a) have focused on aligning these text-based scene graphs with suitable bounding boxes. LSWS (Ye & Kovashka, 2021) utilizes the linguistic structure within triplets for grounding, while another study (Shi et al., 2021) proposes a graph-matching module based on contrastive learning to further improve the grounding performance. **2) Language supervision:** In order to further relax the annotation costs associated with unlocalized scene graphs, the natural language description (i.e., image caption) is used for training the model (Zhong et al., 2021; Kim et al., 2024b; Zhang et al., 2023). SGNLS (Zhong et al., 2021) is the first to enable model training with image captions. LLM4SGG (Kim et al., 2024b) addresses semantic over-simplification and low-density scene graph issues arising from the triplet generation process.

**Weakly Supervised VidSGG (WS-VidSGG).** As the first WS-VidSGG approach, PLA (Chen et al., 2023) utilizes an unlocalized scene graph of the middle frame in a video clip as weak supervision to train a VidSGG model. However, the assumption of a ground-truth scene graph existing in the middle frame among other frames is not only unrealistic but also still requires manual human annotation. On the other hand, our proposed NL-VSGG framework only relies on the readily available video captions to train a VidSGG model, which further reduces annotation costs.

## 5 CONCLUSION

In this work, we propose a weakly supervised **Vid**S**GG** with **N**atural **L**anguage **S**upervision (NL-VSGG) that relieve annotation costs for VidSGG, which is the first time to enable training a VidSGG model with readily available video captions. We identify two key reasons for why a simple adoption of the WS-ImgSGG pipeline to VidSGG fails, i.e., temporality within video captions and variability of action duration. Our Temporality-aware Caption Segmentation (TCS) module captures the temporality within the video captions, while Action Duration Variability-aware Caption-Frame Alignment (ADV) module addresses the variability in the action duration. Furthermore, we propose a novel pseudo-labeling strategy based on motion cues (PLM) to deal with negative action classes in the Action Genome dataset. Our extensive experiments on the Action Genome dataset demonstrate the superiority of NL-VSGG over the simple adoption of WS-ImgSGG pipeline to VidSGG. As a further appeal of NL-VSGG, it allows the VidSGG models to predict a broader range of action classes that are not included in the training data, which makes our proposed framework practical in reality.

However, NL-VSGG has a limitation regarding untrimmed, long-duration videos. For detailed information related to this and the future work addressing it, please refer to the Appendix. R.

## ACKNOWLEDGEMENTS

This work was supported by Institute of Information & communications Technology Planning & Evaluation (IITP) grant funded by the Korea government(MSIT) (RS-2022-II220077, Reasoning, and Inference from Heterogeneous Data), as well as another IITP grant funded by the Korea government(MSIT) (No.2020-0-00004). Furthermore, this research was supported in part by the NAVER-Intel Co-Lab. The work was conducted by KAIST and reviewed by both NAVER and Intel.

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

# Supplementary Material

*- Weakly Supervised Video Scene Graph Generation via
Natural Language Supervision -*

**TASK DESCRIPTION**
In this task, you are given a video caption. Considering the words that indicate the order of events (e.g., then, while, before, and after), your job is to split the given video caption into multiple compositional sentences, and arrange them in chronological order. Note that you should specify the objects for the pronouns used in each of these sentences.

**IN-CONTEXT EXAMPLES**
**Input:** The person is turning on the stove. He then begins to stir some food and after that he picks up a camera and look at it.
**Output:** The person is turning on the stove. → The person stirs some food. → The person picks up a camera. → The person looks at a camera.
**Input:** A person is sitting in bed and texting on a phone while holding a blanket. The person puts the phone down and pulls the blanket up.
**Output:** A person is sitting in a bed. → A person is texting on a phone while holding a blanket. → The person puts the phone down. → The person pulls the blanket up.
**Input:** A person picks up a phone and enters the bathroom through a doorway while talking on the phone. Before walking out of the room, the person puts on shoes and picks up clothes while laughing and dresses them.
**Output:** A person picks up a phone. → A person enters the bathroom through a doorway while talking on the phone. → The person puts on shoes. → The person picks up clothes while laughing. → The person dresses clothes. → The person walks out of the room.
**Input:** A person is sitting on a toilet, picks up a phone and battery that are on the ground, puts the battery into the phone, takes off a jacket, then stands and takes selfies against the bathroom door.
**Output:** A person is sitting on a toilet. → A person picks up a phone and battery that are on the ground. → A person takes off a jacket. → A person stands and takes selfies against the bathroom door.
**Input:** Before taking a drink, a person is undressing, picks up a towel and cleans some glasses.
**Output:** A person is undressing. → A person picks up a towel. → A person cleans some glasses. → A person takes a drink some glasses.
**Input:** Person pulls out phone and begins playing with it then sets it down and pulls the blanket further up.
**Output:** Person pulls out phone. → Person plays with the phone. → Person sets the phone down. → Person pulls the blanket further up.
**Input:** A person watching television and eating a sandwich while laying on the floor and reading book, after a while the person gets up to grab a box.
**Output:** A person watches television and eats a sandwich while laying on the floor. → A person reads a book. → A person gets up to grab a box.
**Input:** A person walks to a pantry, takes out some clothes from it, tosses one on the floor, and puts on another after taking it off again.
**Output:** A person walks to a pantry. → A person takes out some clothes from a pantry. → A person tosses a cloth on the floor. → A person takes a cloth off. → A person puts on a cloth.

**QUESTION OF OUR INTEREST**
**Input:**

Figure 6: Prompt used in the Temporality-aware Caption Segmentation (TCS) module.

## A DETAILS OF PROMPT

In Fig. 6, we provide the complete prompt used in the TCS module, which is discussed in Section 2.2 of the main paper.

## B ABLATION STUDY OF PROMPT DESIGNED FOR COREFERENCE RESOLUTION

In this section, we conduct an experiment to validate the impact of the prompt introduced to address coreference resolution, which aids in specifying the pronouns and extracting accurate triplets. That is, we remove the following sentence in *task description* of Fig. 6: *Note that you should specify the objects for the pronouns used in each of these sentences.*, and evaluate NL-VSGG without it. As shown in Table 5, the performance with the prompt designed for coreference resolution (i.e., Complete Prompt) is superior compared to the performance without that prompt (i.e., w/o Coreference Resolution). It is attributed to the increase in the number of triplets from 58K to 82K (i.e., 40% increase) when using the Complete Prompt, resulting in the alleviation of lack of supervision Kim et al. (2024b). This demonstrates the effectiveness of our prompt design for coreference resolution.

Table 5: Impact of the prompt designed for coreference resolution.

| Method | With Constraint | | No Constraint | |
|---|---|---|---|---|
| | R@20 | R@50 | R@20 | R@50 |
| Complete Prompt | 15.61 | 19.60 | 15.92 | 22.56 |
| +w/o Coreference Resolution | 15.48 | 19.09 | 15.80 | 21.76 |

## C EXPERIMENT ON THE VIDHOI DATASET

**Dataset.** The VidHOI (Chiou et al., 2021) dataset consists of real-life videos capturing daily human activities without a scripted narrative. This dataset includes manually annotated scene graphs on based on keyframes sampled at 1 FPS. Following the processing step of prior studies (Ni et al., 2023; Chiou et al., 2021), we obtain the training and test sets contain 6,366 and 756 videos along with 193,911 and 22,976 frames, respectively. There are 78 object classes and 50 predicate classes, which are divided into 8 spatial classes and 42 action classes. Note that since the VidHOI dataset does not include video captions, we employ a video captioning model (i.e., VideoChat (Li et al., 2023a)) to generate video captions.

**Results.** Table 6 shows performance comparisons under natural language supervision between a simple adoption of WS-ImgSGG pipeline (i.e., WS-ImgSGG) and our proposed framework (i.e., NL-VSGG) on the VidHOI dataset. Beyond the comparisons made within the Action Genome dataset in the main paper, we also observe that in the

Table 6: Results on VidHOI dataset.

| Backbone | Method | Supervision | With Constraint | | No Constraint | |
|---|---|---|---|---|---|---|
| | | | R@20 | R@50 | R@20 | R@50 |
| STTran | Vanilla | Full | 19.48 | 20.30 | 25.63 | 28.05 |
| | +WS-ImgSGG | Weaker | 7.56 | 7.95 | 21.16 | 26.55 |
| | +NL-VSGG | | 10.41 | 10.96 | 21.44 | 27.16 |

VidHOI dataset, NL-VSGG continues to outperform the WS-ImgSGG method, further validating the effectiveness of our proposed framework.

## D    REPLACING CHATGPT AND DAC WITH OTHER MODELS

In this section, we replace the LLM used in the TCS module (i.e., ChatGPT) and the vision-language model used in ADV (i.e., DAC) with alternative models, and use the same prompts shown in Fig. 6 to evaluate the performance. Table 7 shows the results, where Row (a) is the performance of the baseline WS-ImgSGG model, and Row (b) is the performance of the original NL-VSGG reported in the main paper.

Table 7: Performance with other models.

| Row | Module | | With Constraint | | No Constraint | |
|-----|--------|-----|------|------|------|------|
| | TCS | ADV | R@20 | R@50 | R@20 | R@50 |
| (a) | WS-ImgSGG | | 10.01 | 12.83 | 9.02 | 14.05 |
| (b) | ChatGPT | DAC | 15.61 | 19.60 | 15.92 | 22.56 |
| (c) | Mistral-7B | DAC | 14.58 | 18.74 | 14.93 | 21.95 |
| (d) | ChatGPT | InternVideo | 15.58 | 19.66 | 15.92 | 22.38 |

**Replacing an LLM (i.e., ChatGPT) with a smaller LM (i.e., Mistral-7B) in TCS module.** Row (c) shows the performance of replacing ChatGPT-175B OpenAI (2023) with a smaller Mistral-7B Jiang et al. (2023). We observe that although the performance based on the smaller LM is inferior to that of ChatGPT as expected, it still outperforms WS-ImgSGG baseline shown in Row (a). This demonstrates the effectiveness of our proposed NL-VSGG framework.

**Replacing a vision-language model (i.e., DAC) with a video-language model (i.e., InternVideo) in ADV module.** Within the ADV module, we raise the question: is it sufficient to use a vision-language model, which is trained on image-text pair datasets, for aligning segmented sentences with frames while capturing the temporal context between frames? To explore it, we conduct experiments by replacing the image-based vision-language model (i.e., DAC Doveh et al. (2024)) with a video-language model (i.e., InternVideo Wang et al. (2022c)). Specifically, to explicitly reflect temporal context using InternVideo, we group two consecutive frames without overlap and encode each grouped frame to obtain a visual representation, followed by performing Clustering-based Caption-Frame Alignment, as discussed in Section 2.3 of the main paper.

As shown in Table 7, we observe that the performance based on the InternVideo (Row (d)) is competitive with that based on the DAC (Row (b)). It demonstrates that it is sufficient to use a vision-language model for aligning segmented sentences with frames while capturing the temporal context by simply averaging the frames, i.e., centroid in a cluster. This result is consistent with a prior research, i.e., CLIP4CLIP Luo et al. (2022), which demonstrates that simply averaging frame visual representations yields competitive results compared to an advanced model reflecting temporal context between frames.

## E    EXPERIMENT FOR SELECTION STRATEGY OF PSEUDO-LABELING

Here, we conduct experiments over various selection strategies for assigning pseudo-labels of negative classes (i.e., not looking at and not contacting) within unaligned frames, which is mainly addressed in Section 2.2 of the main paper. Specifically, these strategies involve assigning negative classes to the start (S), end (E), or both start and end (SE) frames for each negative class. The experiment results are shown in Fig. 7 reported with R@50 in With Constraint setting. For not looking at, we observe that the performance of assigning it on the end frame is inferior compared to the performance of the start frame ($2^{rd}$ vs. $3^{rd}$ row), while the performance of start+end frames is best ($1^{rd}$ row). It indicates that the start frame for assigning not looking at provides more confident supervision than assignment on the end frame, and assignment on both frames is most beneficial.

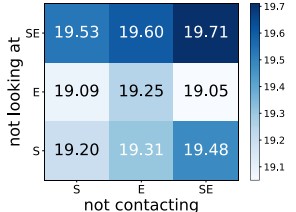

Figure 7: Performance over diverse selection strategies.

For not contacting, in contrast to not looking at, we observe that the performance of assigning it on the start frame is inferior compared to the performance of the end frame ($1^{rd}$ vs $2^{rd}$ column), while the performance of start+end frames is best ($3^{rd}$ column), except when not looking at is exclusively labeled on the end frame ($2^{rd}$ row with $3^{rd}$ column). It indicates that the end frame rather provides more confident supervision than the start frame, while assignment on both frames is generally confident. We attribute the exceptional case to the noisy supervision, where not looking at is only assigned on the end frame.

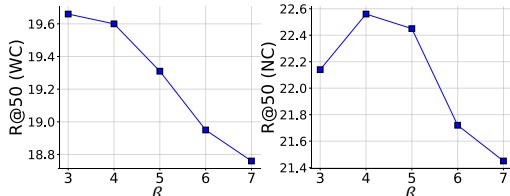 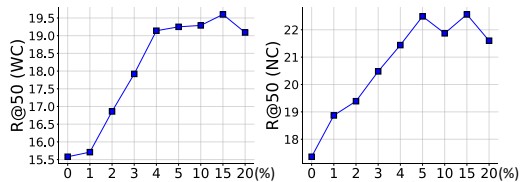

Figure 8: Hyperparameter sensitivity of $\beta$. WC and NC stand for With Constraint and No Constraint setting, respectively.

Figure 9: Hyperparameter sensitivity of $\alpha$. WC and NC stand for With Constraint and No Constraint setting, respectively.

## F  EXPERIMENT FOR DIFFERENT SCENE GRAPH PARSING APPROACH

In this section, we conduct experiments regarding the different approaches for scene graph parsing discussed in Section 2.4 of the main paper. Specifically, there are two approaches to scene graph parsing: one involves extracting triplets and aligning classes with those of our interest based on an LLM Kim et al. (2024b) (LLM-based approach[11]), while the other relies on a rule-based scene

Table 8: Performance over diverse scene graph parsing approaches.

| Scene graph parsing | With Constraint | | No Constraint | |
|---|---|---|---|---|
| | R@20 | R@50 | R@20 | R@50 |
| LLM-based approach | 15.61 | 19.60 | 15.92 | 22.56 |
| SG Parser + KB approach | 11.08 | 14.34 | 11.26 | 16.51 |

graph parser Schuster et al. (2015) with knowledge base-based Miller (1995) alignment (SG parser+KB approach). Note that for the scene graph parsing, these two approaches are widely adopted in the realm of WS-ImgSGG Kim et al. (2024b); Zhong et al. (2021); Zhang et al. (2023); Ye & Kovashka (2021). In Table 8, we observe that the performance of the SG Parser+KB approach is inferior compared to the LLM-based approach. It indicates that the heuristic rule-based parser for triplet extraction from video captions is ineffective, and knowledge base-based alignment struggles to accurately map the classes with complicated action classes (e.g., drinking from, in front of) in VidSGG. In this regard, we demonstrate that the LLM-based approach is particularly effective at WS-VidSGG due to the complex structure of video captions and complicated action classes. To further facilitate research in the WS-VidSGG, we make the triplets extracted based on the LLM publicly available.

## G  EXPERIMENT FOR INTEGRATION OF DYNAMIC RELATIONSHIPS INTO RLIP

In Table 1 of the main paper, we observe that RLIP Yuan et al. (2022) exhibits subpar performance under the zero-shot setting due to its inability to predict dynamic relationships. To explore potential performance enhancement achievable by integrating dynamic relationships into RLIP, we conduct experiments by fine-tuning a pre-trained RLIP on the Action Genome dataset. Furthermore, we ap-

Table 9: Performance of RLIP Yuan et al. (2022) trained with dynamic relationships.

| Model | Supervision | With Constraint | | No Constraint | |
|---|---|---|---|---|---|
| | | R@20 | R@50 | R@20 | R@50 |
| STTran | Full | 33.98 | 36.93 | 36.20 | 48.88 |
| RLIP | Full (Fine-tune) | 31.89 | 36.26 | 34.54 | 41.00 |
| RLIP+STTran | | 34.02 | 40.04 | 35.10 | 42.72 |
| RLIP | Zero-shot | 7.93 | 9.16 | 9.70 | 13.80 |

pend the STTran Cong et al. (2021) module to RLIP in order to facilitate capturing temporal context. As shown in Table 9, we observe that RLIP trained with dynamic relationship substantially boosts performance. Moreover, RLIP with the STTran module further enhances performance by capturing temporal context. It indicates that the incorporation of dynamic relationships is crucial in VidSGG. In summary, we demonstrate the importance of incorporating dynamic relationships and reflecting temporal context in VidSGG.

## H  QUALITATIVE ANALYSIS OF THE ADV MODULE

To qualitatively evaluate the effectiveness of the ADV module, we visualize the results of clustering video frames in the visual space using T-SNE, and show the distribution of their visual-language (VL) scores, i.e., the similarity scores between each segmented sentence and the cluster centroids. In Figure 10, we observe that in the case where a cluster is clearly separated (See A), the VL score distribution is concentrated within that cluster, resulting in the selection of a single cluster. On the

---

[11]In the main paper, we use an LLM-based approach for scene graph parsing.

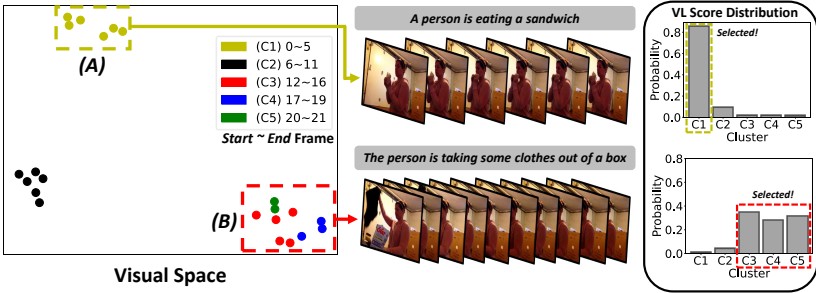

Figure 10: Qualitative analysis of the ADV module.

other hand, in the case where visual features are concentrated but distributed across multiple clusters (See B), the ADV module adaptively selects multiple clusters. This indicates that the ADV module reflects the variability in action duration, allowing it to supervise the model accurately and thereby improve performance.

## I  ABLATION STUDIES ON THE AG DATASET WITHOUT NEGATIVE CLASSES

In Section 3.3 of the main paper, we observed a relatively significant performance improvement in the PLM module due to the dominance of the negative class in the dataset. Therefore, to further clarify the effectiveness of the TCS and ADV modules, we perform ablation studies in which negative classes are excluded from both training and test sets in the Action Genome (AG) dataset. As shown in Table 10, we observe that even in a dataset where the PLM module cannot be applied, TCS and ADV modules still show superiority, proving their effectiveness.

Table 10: Ablation studies where negative classes are excluded from the AG dataset.

| TCS | ADV | With Constraint | | No Constraint | | Mean | |
|-----|-----|------|------|------|------|------|------|
| | | R@20 | R@50 | R@20 | R@50 | R@20 | R@50 |
| | | 12.98 | 17.10 | 13.28 | 19.38 | 13.13 | 18.24 |
| ✓ | | 15.34 | 20.24 | 15.47 | 21.72 | 15.41 | 20.98 |
| ✓ | ✓ | **15.95** | **20.74** | **16.19** | **23.04** | **16.07** | **21.89** |

## J  HYPERPARAMETER SENSITIVITY

In NL-VSGG, two hyperparameters are used: $\beta$ for defining $K$ (i.e., $\frac{|V|}{\beta}$) in ADV module (Section 2.3), and $\alpha$ for assigning negative classes within unaligned frames in PLM module (Section 2.5). We analyze the sensitivity of these hyperparameters.

**Hyperparameter $\beta$ for ADV module.**  In Fig. 8, we conduct experiments over various $\beta$s. We observe that the performance decreases with large $\beta$ which reduces the cluster number $K$. It indicates that large $\beta$ (i.e., small $K$) cannot capture the fine-grained variability of action duration, resulting in deteriorating performance. On the other hand, small $\beta$ (i.e., large $K$) helps to capture fine-grained variability of action duration, leading to increasing performance. However, rather small $\beta$ (i.e., 3) decreases performance in No Constraint setting. We attribute it to the fact that rather small $\beta$ divides the frames into overly fine-grained clusters, making it difficult for the vision-language model to distinguish highly discriminative clusters in terms of similarity scores. In this regard, it is beneficial to appropriately set $\beta$ as 4 to capture the variability of action duration.

**Hyperparameter $\alpha$ for PLM module.**  In Fig. 9, we conduct experiment over various $\alpha$s. We observe that the performance consistently increases up to 5%, followed by fluctuation beyond 5%. It is worthwhile noting that the pseudo-labels with smaller $\alpha$ would provide more confident supervision to models since their associated subject and object are distinctly getting farther over time. It suggests that up to 5%, clear supervision is provided for negative classes within unaligned frames, but beyond that ratio, noisy supervision is included. Hence, setting $\alpha$ at around 5% is preferable. However, we opt for a value of $\alpha$ at 15% as it marginally enhances performance under the With Constraint setting.

## K  DETAILS OF EXPANDED ACTION CLASSES

In Table. 14, we enumerate all the expanded action classes sorted by frequency in descending order.

Table 11: Summarization of cost for an LLM.

| Module | Num. Output/Input tokens per video | Cost per video |
|--------|-----------------------------------|----------------|
| TCS | 0.045k / 0.68k | $0.00041 |

## L  COST FOR UTILIZATION OF A LARGE LANGUAGE MODEL

For the TCS module, the cost of using ChatGPT OpenAI (2023) is summarized in Table 11. Given that the cost of input tokens and output tokens is $0.5 and $1.5 per 1M tokens, respectively, the cost per video is computed by $(680/1M) \times 0.5 + (45/1M) \times 1.5$.

## M  EXPERIMENT FOR DIFFERENT CLUSTERING IN THE ADV MODULE

To investigate the impact of different clustering strategies within the ADV module, we conducted experiments where we replaced the K-means clustering strategy with Agglomerative clustering and Gaussian Mixture Model (GMM) clustering

Table 12: Performance over various clustering strategies.

| Clustering Algorithm | With Constraint | | No Constraint | | Mean |
|----------------------|-----------------|---------|---------------|---------|------|
| | R@20 | R@50 | R@20 | R@50 | |
| K-Means | 15.61 | 19.60 | 15.92 | 22.56 | 18.24 |
| Agglormerative | **15.78** | 19.69 | **16.12** | 23.01 | 18.65 |
| GMM | 15.31 | **19.80** | 15.85 | **23.93** | **18.72** |

strategies. As shown in Table 12, we observed that other clustering strategies exhibit competitive performance compared to that of the K-means clustering strategy, indicating that our proposed framework is robust to other clustering strategies. Another observation is that the performance with the GMM clustering strategy is relatively better on average. This result aligns with previous studies (Huang et al., 2022; Zheng et al., 2022) that assume proposal distributions as Gaussian in the temporal grounding task in that GMM also assumes the same thing, resulting in effective clustering within the ADV module.

## N  EXPERIMENT COMBINING WEAKLY SUPERVISED AND FULLY SUPERVISED DATASETS

In this section, we investigate a scenario where our proposed approach meets with the fully supervised approach. To this end, we attempted to fine-tune a model, which was initially trained on a weakly supervised dataset (i.e., pseudo-localized scene graphs in Section 2.5), to the fully super-

Table 13: Performance for a combination of weakly supervised and fully supervised data.

| Training dataset | With Constraint | | No Constraint | | Mean |
|------------------|-----------------|---------|---------------|---------|------|
| | R@20 | R@50 | R@20 | R@50 | |
| AG (Full) | 33.98 | 36.93 | 36.20 | 48.88 | 39.00 |
| AG+MSVD (Weak) → AG (Full) | **34.84** | **37.64** | **38.40** | **49.55** | **40.11** |

vised dataset, i.e., ground-truth localized scene graphs. In this process, we assumed that a model trained on a larger dataset would provide more effective weight initialization, leading us to leverage the model trained on both the AG caption and MSVD caption datasets, as detailed in Section 3.6 of the main paper. Interestingly, as shown in Table 13, we observed that the performance of the model fine-tuned to the AG dataset significantly outperformed that of the model initially trained on the AG dataset, implying that our proposed framework can synergize with the fully supervised approach.

## O  DISCUSSION OF IMPROVEMENT FOR PARSING AND GROUNDING

For future work, we discuss the techniques developed for scene graph parsing and grounding, detailed in Section 2.4 of the main paper.

**Improvement for Scene Graph Parsing.** We may employ the ensemble approach (Wang et al., 2022b) to extract more high-quality scene graphs. Specifically, for an LLM-based parser, we can utilize temperature sampling or top-k sampling to extract diverse triplets for each segmented sentence, followed by taking a majority vote over diverse triplets to extract the most consistent triplet. With this technique, we can further develop the scene graph parsing over the state-of-the-art method.

**Improvement for Scene Graph Grounding.** We may improve the grounding accuracy. Basically, a triplet is grounded to a bounding box when the bounding box's class matches the object class of the triplet parsed from the caption. However, inherent challenges in videos such as motion blur, fast

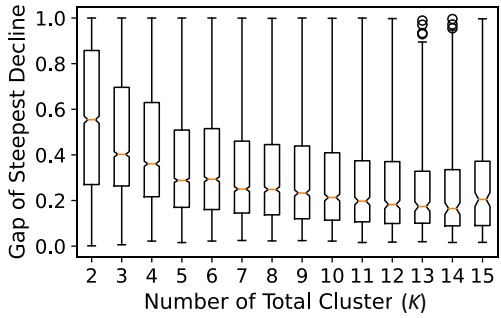
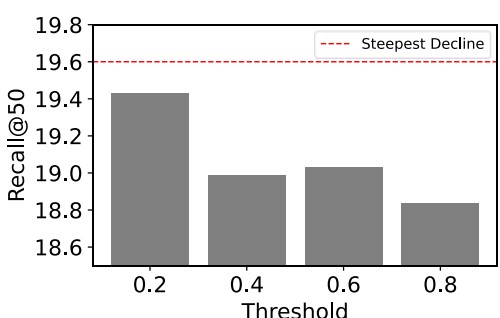

Figure 11: The gap in the similarity score at the point of the steepest decline over the number of total clusters (i.e., $K$), which is proportional to the frames.

Figure 12: Performance across various fixed thresholds in the ADV module, conducted under the With Constraint setting.

movement, and occlusion often hinder accurate object class classification within the bounding box, resulting in grounding failures. To address it, we can utilize adjacent frames successfully grounded. Specifically, in the case where the object class of the bounding box in the target frame is ambiguous so that grounding fails, we can ensure grounding by selecting a bounding box with high IoU and visual similarity to a bounding box of an object that is grounded in an adjacent frame. This technique could compensate for the failures in the target frame, ensuring more reliable grounding.

## P    DISCUSSION OF THE STEEPEST DECLINE IN THE ADV MODULE

In the ADV module, detailed in the Section 2.3 of the main paper, we originally determined the relevant clusters depending on the steepest decline in the vision-language similarity score. Here, we explore the sensitivity of the steepest decline criterion to noise in the similarity score. To this end, we analyze the gap in the similarity score at the point of steepest decline[12] over the total number of clusters (i.e., $K$), which is proportional to the video frames. As shown in Figure 11, we observe that the gap in the steepest decline decreases as the number of total clusters increases (from the left to the right on the x-axis). This is because the similarity score is distributed across a greater number of clusters, thereby reducing the gap. However, it is important to note that from eight clusters onward, the gap converges to 0.2, demonstrating that even with noise in the similarity score, the steepest decline approach still reliably selects relevant clusters using this relatively large 0.2 gap.

In this regard, we raise a question: Why not determine the relevant clusters through a point where the gap in the decline exceeds 0.2 instead of using the steepest decline? Specifically, we could sort the similarity scores in descending order, find the first point where the gap exceeds 0.2, starting from the highest score, and select all the clusters preceding this point. Therefore, we conducted additional experiments where we set the threshold as a hyperparameter, ranging from 0.2 to 0.8, to determine the relevant clusters through a point where the gap in the decline exceeds this threshold. As shown in Figure 12, we had two observations: 1) Setting the threshold at 0.2 shows competitive performance with the steepest decline, aligning with an aforementioned discussion that a 0.2 gap is robust to determine the relevant clusters. 2) As the threshold increases, we observe a decrease in performance. This is because more irrelevant clusters are assigned as the threshold increases, thereby introducing noise.

However, we argue that *this thresholding approach requires meticulous analysis of the gap in the steepest decline for each dataset to set the threshold, whereas the steepest decline approach has the advantage of adaptively capturing relevant clusters without needing to analysis each dataset.*

---

[12]For example, in Figure 10 of the Appendix, the gap in the similarity score at the point of the steepest decline is C1-C2 for ($A$), and C4-C2 for ($B$).

## Q  ADDITIONAL RELATED WORK

### Q.1  LARGE LANGUAGE MODEL-BASED SCENE GRAPH GENERATION

As Large Language Models (LLMs) got a surge of interest from various domains, LLMs have also been applied to the SGG task to leverage the rich semantic knowledge. RECODE (Li et al., 2024) utilizes LLMs to generate an informative description for a predicate, enabling it to capture the fine-grained visual cues. For zero-shot scene graph generation task, ELEGANT (Zhao & Xu, 2023) extracts the candidate relationships generated by the LLMs, which have profound reasoning and commonsense knowledge. For weakly supervised SGG, LLM4SGG (Kim et al., 2024b) alleviates the long-tailed predicate issue and scarcity of datasets via the LLMs. In this work, we leverage the LLMs to understand the temporality within video captions for the WS-VidSGG task.

## R  FUTURE WORK

A potential limitation of our work is that NL-VSGG is mainly designed for handling relatively short-length videos (∼120 seconds). As future work, we plan to generalize NL-VSGG to longer untrimmed videos. One possible direction would be to perform untrimmed video temporal localization (Wang et al., 2023; Yan et al., 2023; Li et al., 2023b) tasks in computer vision, wherein short-length clips are extracted from longer videos, and apply NL-VSGG to each clip for training a VidSGG model. This approach would enable the training of the VidSGG model with untrimmed longer videos.

Another direction for future work is to explore the possibility of using Multimodal Large Language Models (MLLMs) to extend open-set or multi-label prediction in the VidSGG task. Specifically, for open-set prediction, we can query the temporal relationship between a subject and an object to the MLLMs by inserting the union box and previous frames' caption for temporal context. For multi-label prediction, temperature (Ficler & Goldberg, 2017) or top-$k$ sampling (Radford et al., 2019), commonly used in the NLP for generating diverse answers, can be used to generate various temporal relationships.

Table 14: Enumeration of all expanded action classes.

put, pick up, open, take, hold, sit on, eat, grab, close, look at, throw, take off, drink,
walk into, sit in, place, on, sit at, watch, with, put on, pour, walk to, read, set,
drink from, in, clean, carry, walk over to, stand in, walk through, turn on, fold,
work on, take out, look out, sit, remove, play with, use, get, look in, lie on,
sweep, talk on, from, wash, fix, sit down on, tidy up, enter, walk out of, leave, into,
move, turn off, lay on, walk in, go to, out of, look into, walk down, at, vacuum, play on,
sit down, stand in front of, cook, turn, look through, off, stand on, wipe, run into,
have, snuggle with, drop, walk up, make, lay down on, stand at, in front of, tidy,
walk around, opening, check, do, pull out, toss, adjust, of, put down, pull, stand by,
walk out, sit down at, shut, touch, open up, get up from, sit down in, pick, lie in,
walk, hug, snuggle, lay in, onto, type on, look, walk across, drink out of, set down,
sleep on, to, come into, flip through, write in, laugh at, walk up to, go through,
stand, lay, lie down on, sneeze into, put away, talk to, wrap in, stir, sneeze, dry, wear,
start, write, wrap, down on, reach for, stand next to, next to, hang, play, go into,
stare at, retrieve, get out of, straighten, eat from, through, run through, run, grasps,
walk towards, off of, smile at, under, look inside, answer, cook on, clean up, go over to,
exit, fill, run out of, laugh, wipe off, stand near, look out of, reach into, walk over,
smile, lean against, keep, cover with, tie, sweep with, rearrange, write on, stand hold,
back on, sit in front of, mess with, wipe down, hold onto, hang up, organize, grasp,
examine, prepare, run down, go out, wave, go back to, find, shake, walk in with,
cover, bring, inspect, come in, undress, arrange, down, inside, awaken on, run to,
push, dust, cover in, empty, lean on, sip from, by, stack, texte on, straighten up,
walk with, interact with, walk away with, on the floor, bend down, lay down, flip,
kick off, cuddle with, stand up, stand up from, clean off, get dress, on top of,
leave through, walk away from, lock, fiddle with, reach, take out of, hold up, dump,
go up, hand, fold up, point at, run around, change, cuddle, run in, kick, fluff,
walk away, stand behind, knock, pour out, over, come through, swing, rub,
enter through, get up out of, walk in through, unfold, unlock, sneeze on, scrub,
take picture of, bend over, button up, walk past, type, consume, look for, continue,
take a bite of, lie down, see, button, getting dress, near, smell, hit, get up off,
lay down with, sit down to, take picture with, move to, get out, zips up, lift, turn out,
dress in, walk toward, talk, give, rinse, look up at, read from, sit with, run up, get into,
stop at, wrap up in, get off, sleep in, show, cover up with, twirl, wrap around, gather,
lie down in, proceed to, be, around, check out, view, tidy up with, spray,
undress out of, walk in hold, come in through, eat out of, hold on to, approach,
appear to be in, swallow, fall on, sit next to, kneel on, dusting, stand nearby, walk by,
search for, snuggle in, get on, stop, leave with, get up, for, search through, sniff,
sit down in front of, on the ground, sweep up, undress from, back in, sit down with,
asleep on, walk from, lean, look around, fidget with, look over, reach over, run out,
cook at, rummage through, back into, polish, dress, come out of, dig through, return to,
wake up from, swinge, comb, brush, sort, return, talk with, do work on, ball up,
move towards, climb up, smile into, sit up, walk in front of, awaken, pace in, seize,
undress in front of, climb into, roll up, finish, wet, pull up, place on, do something on,
stand up with, spread, admire, out, wrap themselves in, untie, seat at, spill, wake up on,
stand with, tap on, go in, go, juggle, behind, awaken from, work at, turn to,
dance around, shine, turn away from, up to, pat, flick, zips, clean with, pack, reach over to,
turn back to, walk back to, knock over, exit through, return with, walk back out of,
vacuum around, run across, walk back down, full of, walk around with, follow, go out of,
plug in, stand watch, away from, bite, switch, dump out, rock in, sprinkle, stick,
sip out of, sit back down on, prop, underneath, seat on, cook with, do something with,
recline on, flip on, lie, seat in, wake up, replace, continue up, reach in, kneel down,
unbutton, text on, clothe, come to, appear to be tidy up, tie up, fill up, climb,
reach up on, climb on, lay down in, down onto, on the back of, face, move from, unpack,
on their lap, walk out with, go and sit on, walk back into, get to, look down at, drape,
sleep at, straighten out, rifle through, move around, appear, shake out, proceed to eat, sit by,
take from, pace back and forth, walk back, film, shuffle, come back in, up on, back onto,
enjoy, their, exercise, eat in, tap, sit down to watch, on the shelf, dust off, walk back across

