# OpenReview forum: "Weakly Supervised Video Scene Graph Generation via Natural Language Supervision"
_ICLR.cc/2025/Conference — ICLR 2025 Poster_

### Official Review · Reviewer_61fs · 2024-11-02

**Soundness:** 3
**Presentation:** 3
**Contribution:** 2
**Rating:** 6
**Confidence:** 4

**Summary:**

This manuscript tackles the problem of extracting scene graphs from video inputs by only using raw video captions during the training stage. The proposed method focuses on extracting pseudo training data from the raw captions to be consumable by supervised VidSGG models. The proposed method consists of TCS (temporally sorting the atomic events in the complex captions), ADV (handling the variable length of atomic actions), and k-means clustering with some smoothings to assign clusters of frames to each of the atomic actions.

**Strengths:**

VidSGG is an important problem, and IMO has been overlooked. I believe this line of research is very important, and I believe that authors selected a good approach by utilizing only captions to train a vidsgg.

Also, the temporal inconsistency and variability of action lengths are two main difficulties for any video-language tasks. I believe that this paper systematically decomposed them and addressed them each separately.

**Weaknesses:**

The main weakness of this manuscript is that it does not contribute to any core learning methods. For example, no visual understanding or natural language understanding method has been improved. It is mostly a wise tooling over LLMs and some basic approaches like clustering. Even the final training has been delegated to off-the-shelf vidsgg methods.

**Questions:**

1- In Table 1, what about combining the full supervision and weak supervision data to see if the weak supervision can give a boost to supervised results? The combination could be combining the training data or doing pretraining on weak supervision and then fine-tuning on fully supervised data. IMO, we can always have some supervised data, and the value of this work will shine if we see that it can boost the results on top of that.

2- Lines 240-244: I think this part needs more clarification. A) Does this filtering contribute to the final results? B) In the given example, will I^1 be assigned to S1, or will it not be assigned to either S1 or S2? C) Videos can be formed of different shots that can jump between two actions back and forth. How those cases will be handled?

3- Do the clustering steps have any other purpose than computation efficiency? Why not compare each singleton frame and the Segmented Sentences?

4- I would like to see the examples that have been passed to the LLM. A) How are they selected? B) How does the quality of output change by changing those examples? I see a reproducibility risk here.

---

> ### Author Response · Authors · 2024-11-22
> **(1/N) Author Rebuttal**
>
> We thank the reviewer for the constructive feedback on our paper!
>
> **W. No Introduction of Core Learning**
> We agree that our proposed method does not introduce any core learning methods. However, we hope the reviewer will recognize *our contribution in proposing a novel pipeline that enables the training of a VidSGG model only with video captions for the first time*, despite our method not being equipped with a core learning method. In the line of weakly supervised ImgSGG, [1] utilized tools like WordNet and Scene Parser to train the ImgSGG model using only image captions, thus paving the road for a weakly supervised approach. Similarly, our work could serve as a cornerstone to explore the potential of training the VidSGG model using video captions, and we expect it to be further enhanced through diverse learning methods in the future.
>
> [1] Learning to Generate Scene Graph from Natural Language Supervision. Zhong et al. ICCV’21
>
> **Q1. Combination of Weakly Supervised and Fully Supervised Datasets**
> |          Training dataset          | W/C |       | N/C |       |  Mean |
> |:----------------------------------:|:---------------:|:-----:|:-------------:|:-----:|:-----:|
> |                                    |       R@20      |  R@50 |      R@20     |  R@50 |       |
> |              AG (Full)             |      33.98      | 36.93 |     36.20     | 48.88 | 39.00 |
> | AG+MSVD Caption (Weak) → AG (Full) |      34.84      | 37.64 |     38.40     | 49.55 | 40.11 |
> ###### *W/C: With Constraint, N/C: No Constraint
>
> As the reviewer suggests, our work will indeed shine if we demonstrate the effectiveness of the combination of the weakly supervised data and the fully supervised data. To this end, we attempted to finetune the model, which was initially trained on weakly supervised data, on the fully supervised data, i.e., the Action Genome (AG) dataset. In this process, we assume that a model trained on a larger dataset would provide more effective weight initialization, leading us to leverage the model trained on both the AG caption and MSVD caption datasets, as detailed in Section 3.4. Interestingly, as shown in the table, we observed that the performance of the model finetuned on the AG dataset in a supervised manner significantly outperformed that of the model initially trained on the AG dataset, implying that our proposed method can synergize with the fully supervised approach. We sincerely thank the reviewer for the insightful suggestion, which has enhanced the value of our work. We have incorporated it into Appendix O to further improve the quality of our work.
>
> **Q2. Clarification of Removing Unrealistic Frame**
> |                 Method                | W/C |       | N/C |       |
> |:-------------------------------------:|:---------------:|:-----:|:-------------:|:-----:|
> |                                       |       R@20      |  R@50 |      R@20     |  R@50 |
> | VSNLS                                 |      15.61      | 19.60 |     15.92     | 22.56 |
> | VSNLS w/o Removing Unrealistic Frames |      15.28      | 19.27 |     15.52     | 21.86 |
> ###### *W/C: With Constraint, N/C: No Constraint
> We clarify the part “Removing Unrealistic Caption-Frame Alignment” one by one.
> (A) Regarding the contribution to the final results, we performed an ablation study of retaining the unrealistic frames (w/o removing unrealistic frames), which might erroneously supervise the model. As shown in the table, we observe that this incorrect supervision deteriorates the performance, demonstrating the effectiveness of removing the unrealistic frame.
>
> (B) Regarding the assignment of frame $I^1$ in the example, $I^1$ will not be assigned to $S^1$. To further clarify, frame $I^1$ has a direct relationship with segmented sentence $S^2$, but not with segmented sentence $S^1$. Therefore, aligning frame $I^1$ to segmented sentence $S^1$ would result in additional noise. That’s why we do not assign frame $I^1$ to segmented sentence $S^1$.
>
> (C) Regarding cases where the different shots jump back and forth between two actions, we can handle this by closely analyzing the vision-language score in the ADV module. The vision-language score, computed for alignment between segmented sentences and frames, would appear high in several non-consecutive shots, suggesting that the same action occurs in several shots. Based on this intuition, we can decide whether frames that violate the temporal order are unrealistic or occur during back-and-forth actions by counting those unrealistic cases. To align the segmented sentence to the frames more accurately,  we could consider the violated frames as unrealistic and remove them if there are few cases (i.e., 1~2), while considering them as realistic and retain them if such cases are frequent (i.e., 3 or more), indicating two actions jumping back and forth.

---

> ### Author Response · Authors · 2024-11-22
> **(N/N) Author Rebuttal**
>
> **Q3. Explanation for Clustering Purpose and Simple Approach**
> We begin by clarifying the purpose of the clustering step.  For K-means clustering, we aim to generate multiple proposals [2], each of which has distinct semantics with respect to action type, thereby grouping frames into different clusters. However, it is important to note that some clusters might have background semantics, meaning that they do not involve any specific actions or they fall within the interval between two actions. Following the clustering process, we compare the semantics of the cluster with those of segmented sentences, leading the cluster with background semantics to be not aligned with any of the sentences, while the cluster showing an action to be aligned. In this context, as per the reviewer’s suggestion, if we compare each singleton frame and the segmented sentences one by one and assign each frame to the segmented sentences, this approach would inevitably assign frames with background semantics to segmented sentences, which is not expected.
>
> [2] Zero-shot Video Moment Retrieval from Frozen Vision-Language Models. Luo et al. WACV’24
>
> **Q4. Explanation for Selection of Examples and Impact of Changing them on Quality**
> In Figure 6 of Appendix A, we have presented 8 examples that have been passed to the LLM in our experiments. The same 8 examples are used for every video in our dataset. Our intention was to evenly include diverse temporal markers (e.g., after, before, then, and while) as much as possible using the Action Genome caption dataset, aiming to enhance the LLM’s understanding of temporal markers.
>
> We agree with the reviewer’s concern about reproducibility when changing the examples passed to the LLM. Therefore, we conducted a qualitative analysis to compare 200 outputs of the TCS module (i.e., segmented sentences) using the original examples shown in Figure 6 of Appendix A with those obtained from randomly sampled examples (repeated three times). Since this comparison requires intensive human labor, we utilize a scene parser to convert each segmented sentence into triplets (i.e., subject-predicate-object) and perform an exact match for each result against the original examples’ output. We noted a 92.50% accuracy with a 2.63 standard deviation, verifying the robustness of the TCS module despite the change of examples.

---

> ### Comment · Reviewer_61fs · 2024-11-26
> **Thanks for the responses**
>
> Thanks for all the clarifications, more specifically for doing the extra analysis for the Q1.

---

> ### Author Response · Authors · 2024-11-27
>
> We are grateful for the valuable feedback provided by the reviewer. Your insightful comments have greatly enhanced the quality of our work.

---

### Official Review · Reviewer_6GYf · 2024-11-03

**Soundness:** 4
**Presentation:** 4
**Contribution:** 3
**Rating:** 6
**Confidence:** 3

**Summary:**

This paper conducts video scene graph generation in a weakly supervised manner. Several key steps are employed to generate scene graph labels from video-text annotations. The authors first design text prompts for multimodal large language models to parse the video captions into sentences along a time horizon. Then, a clustering-based caption-frame alignment framework is designed to align video frames with the segmented sentences. With images and their aligned texts, the authors use existing scene graph parsing methods and scene grounding methods to generate scene graphs. The authors also conduct experiments on the video scene graph generation benchmark , the Action Genome (AG) dataset, to validate the effectiveness the proposed method.

**Strengths:**

1. This paper proposes a weakly supervised pipeline for video scene graph generation, an important research topic in multimodal learning. Researchers can build a larger scene graph generation dataset with this pipeline from annotated video-text pairs. Researchers can also leverage the MLLMs to generate scene graphs without any human labels in further days.
2. The design of the temporality-aware caption segmentation module is interesting; it can convert the video's text caption into multiple sentence segments. MLLM may find it challenging to finish such tasks. It would be helpful to add more details about the segment process.
3. Aligning the segmented sentences with the video frames is critical for generating a scene graph. The authors provide many details about conducting the clustering-based alignment.

**Weaknesses:**

Although the proposed VSNLS pipeline can achieve good results on VidSGG tasks, its innovation still needs to be enhanced and emphasized. The VSNLS pipeline has four critical steps: the temporality-aware caption segmentation part, the action duration variability-aware caption-frame alignment part, the generation of pseudo-localized scene graphs part, and the adverse action classes generation part. Except for the action duration variability-aware caption-frame alignment part, I am unclear about the novelty and technical insights in other sections.
1. In section 2.2, the authors design prompts to segment the video captions into sentences. Since there are a lot of prompt engineering works, what are the new insights in this part that can motivate the following researchers to conduct their work? Are there some key points that need to be noticed in design prompts?
2. In section 2.4, the scene graph parsing and grounding utilize state-of-the-art methods to generate graph nodes. I guess readers are more willing to see how the authors develop some techniques to make these state-of-the-art methods conduct parsing or grounding more accurately.
3. The experimental setting is organized in a bit of chaos. How the training, validation, and testing sets are split is not introduced. From line 316 to line 325. the reader can hardly understand whether the model is trained on the AG dataset or the combination of the AG caption and the MSVD caption dataset.
4. There is no ablation study on the effectiveness of the microdesign in the ADV part. How different clustering strategies influence the final results is important to understand the rationale behind the VSNLS framework.

**Questions:**

I have no further questions. Please see the weakness part.

---

> ### Author Response · Authors · 2024-11-22
> **(1/N) Author Rebuttal**
>
> We appreciate the reviewer's constructive feedback on our paper!
>
> **W1. Key Point and New Insight of TCS module**
> The key point in the TCS module is the instruction prompt (i.e., Considering the words ~ chronological order), which guides the chronological segmentation of captions. Another key point is the prompt resolving the coreference issue, stated as “Note that you should specify the objects for the pronouns used in each of these sentences.” Actually, we observed that the performance decreases when the prompt tailored to address coreference issues is omitted, as detailed in Appendix B.
>
> Above all, it is important to note that even if our designed prompt may seem straightforward and not give new insight into the field of prompt engineering, we believe that *the effectiveness of our prompt gives a new insight regarding the necessity of segmenting captions in chronological order for the weakly supervised video scene graph generation task, and that this could be addressed via LLMs*. We kindly request the reviewer to see our work from a broader perspective, acknowledging that our proposed framework enables training the VidSGG using only video captions for the first time.
>
> **W2. Development of Techniques for Parser and Grounding**
> Regarding the enhancement of scene graph parsing, we may employ the ensemble approach [1] to extract more high-quality scene graphs. Specifically, for an LLM-based parser, we can utilize temperature sampling or top-k sampling to extract diverse triplets for each segmented sentence, followed by taking a majority vote over diverse triplets to extract the most consistent triplet. With this technique, we can further develop the scene graph parsing over the state-of-the-art method.
>
> Regarding the enhancement of scene graph grounding, we may improve the grounding accuracy. Basically, a triplet is grounded to a bounding box when the bounding box’s class matches the object class of the triplet parsed from the caption. However, inherent challenges in videos such as motion blur, fast movement, and occlusion often hinder accurate object class classification within the bounding box, resulting in grounding failures. To address it, we can utilize adjacent frames successfully grounded. Specifically, in the case where the object class of the bounding box in the target frame is ambiguous so that grounding fails, we can ensure grounding by selecting a bounding box with high IoU and visual similarity to a bounding box of an object that is grounded in an adjacent frame. This technique could compensate for the failures in the target frame, ensuring more reliable grounding.
>
> We believe that these techniques would facilitate future work for weakly supervised video scene graph generation. Therefore, we have included it in Appendix P of the revised manuscript.
>
> [1] Self-Consistency Improves Chain of Thoughts Reasoning in Language Models. Wang et al. ICLR’23
>
> **W3. Clarification of Experimental Setting**
> To clarify the experimental setting, we have revised the Dataset section (Section 3.1) in the manuscript. To summarize, as stated in Line 312 and 325, we utilized 7,454 and 1,747 videos from the Action Genome dataset for training and evaluation, respectively. Furthermore, for model training clarification, we utilized the Action Genome caption dataset for analysis throughout our paper. On the other hand, the MSVD caption dataset is only used in Section 3.4. We apologize for any confusion regarding the experimental setting and would appreciate your review of the revised Dataset section in Section 3.1.

---

> ### Author Response · Authors · 2024-11-22
> **(N/N) Author Rebuttal**
>
> **W4. Result of Different Clustering**
> | Clustering Strategy | W/C|       | N/C|       |  Mean |
> |:-------------------:|:---------------:|:-----:|:-------------:|:-----:|:-----:|
> |                     |       R@20      |  R@50 |      R@20     |  R@50 |       |
> | K-Means             |      15.61      | 19.60 |     15.92     | 22.56 | 18.42 |
> | Agglomerative       |      15.78      | 19.69 |     16.12     | 23.01 | 18.65 |
> | GMM                 |      15.31      | 19.80 |     15.85     | 23.93 | 18.72 |
> ###### *W/C: With Constraint, N/C: No Constraint
> To investigate the impact of different clustering strategies within the ADV module, we conducted experiments where we replaced the K-means clustering strategy with Agglomerative clustering and Gaussian Mixture Model (GMM) clustering strategies. As shown in the table, we observed that other clustering strategies exhibit competitive performance compared to that of the K-means clustering strategy, indicating that our proposed framework is robust to other clustering strategies. Another observation is that the performance with the GMM clustering strategy is relatively better on average. This result aligns with previous works [2,3] that assume proposal distributions as Gaussian in the temporal grounding task in that GMM clustering also assumes the same thing, resulting in effective clustering within the ADV module.
> We thank the reviewer for providing a valuable suggestion to conduct a fine-grained analysis of our proposed framework. To improve the quality of our work, we have incorporated this analysis in Appendix N.
>
> [2] Weakly Supervised Temporal Action Localization via Representative Snippet Knowledge Propagation. Huang et al. CVPR’22
> [3] Weakly Supervised Temporal Sentence Grounding with Gaussian-based Contrastive Proposal Learning. Zheng et al. CVPR’22

---

> > ### Comment · Reviewer_6GYf · 2024-11-26
> > **Relpy to the rebuttal**
> >
> > After reading the authors' responses, most of my concerns have been resolved. I choose to maintain my rating.

---

> ### Author Response · Authors · 2024-11-27
>
> We sincerely appreciate the reviewer's valuable feedback, which has contributed to the further development of our paper.

---

### Official Review · Reviewer_aNnm · 2024-11-04

**Soundness:** 2
**Presentation:** 2
**Contribution:** 2
**Rating:** 6
**Confidence:** 5

**Summary:**

This paper introduces a framework, VSNLS, for Video Scene Graph Generation (VidSGG) that leverages natural language supervision from video captions to reduce the high cost of manual annotation. Unlike existing methods, VSNLS uses two modules tailored for video data: a Temporality-aware Caption Segmentation (TCS) module to capture time markers in captions, and an Action Duration Variability-aware Caption-Frame Alignment (ADV) module to align captions with frames based on action duration. This approach enables the model to learn from weak supervision, allowing it to predict dynamic relationships and even generalize to unseen actions. The proposed method is shown to be effective on the Action Genome dataset, demonstrating improved performance over traditional image-based weak supervision techniques.

**Strengths:**

VSNLS addresses the high annotation cost of fully supervised Video Scene Graph Generation (VidSGG) by using weak supervision from video captions. This method eliminates the need for extensive manual annotation of all frames, which is both time-consuming and costly.

**Weaknesses:**

-Dependency on Co-occurrence Priors: VSNLS, like many video-based scene graph models, often relies on co-occurrence priors to estimate relationships. This dependency can reduce the generalizability of the model to unseen object combinations, potentially limiting its performance on diverse or novel datasets where the objects and relations deviate from the training distribution.

-Handling Long-term Relations: Clip-based methods in VSNLS might struggle with capturing relations that unfold over extended sequences, which require a larger receptive field. If clip length is limited (e.g., fixed at 30 frames), the model may miss detecting relations that span longer temporal windows, thereby impacting its ability to accurately model long-term dependencies.

-Occlusion and Tracking Limitations: Although clip-based approaches are often advocated for mitigating long-term tracking issues like occlusion, the use of short-term tubelets might still face challenges in fragmented tracking. Occlusions or visual artifacts can still lead to lost tracklets or misalignments, reducing the reliability of relationship detection over long video durations.

-High Computational Cost for Short Clips: Analyzing shorter clip sequences may require repeated processing of overlapping frames to maintain temporal context, leading to higher computational and memory demands. This could make the model less efficient and limit scalability for very long video sequences where full contextual understanding is crucial.

-Important References Missing:
[1] Video visual relation detection via iterative inference. ACM MM 2021.
[2] Winner: Weakly-supervised hierarchical decomposition and alignment for spatio-temporal video grounding. CVPR 2023.
[3] In Defense of Clip-based Video Relation Detection. TIP 2024.

**Questions:**

The results are reported on the Action Genome dataset, but it is unclear how well the framework would perform on other VidSGG datasets with different types of actions or domain-specific captions. How about VidOR dataset?

How does VSNLS handle noisy or ambiguous captions? Since the method depends on the quality of video captions, it would be helpful to understand how robust the framework is to variations in caption detail and accuracy.

---

> ### Author Response · Authors · 2024-11-22
> **(1/N) Author Rebuttal**
>
> We are sincerely grateful for the reviewer's insightful feedback. We have carefully addressed all the reviewer’s concerns. Before proceeding, we’d like to emphasize that this work follows *frame-level VidSGG* rather than video-level VidSGG. We would appreciate it if you could take this distinction into account when reviewing the response accordingly.
>
> **W1. Dependency of Co-occurrence Priors**
> We agree that using the co-occurrence prior to estimate relationships can deteriorate the generalizability of the model to unseen triplets, as supported by [1]. However, we’d like to clarify the utilization of co-occurrence priors in VSNLS. Unlike existing work in VidSGG [2] that utilizes co-occurrence priors for relationship estimation to improve overall performance, *we do not rely on co-occurrence priors for relationship estimation*. In fact, our approach could potentially enhance the model’s generalizability to unseen triplets by leveraging video captions, which comprise a broader range of diverse compositional relationships. Please note that using captions for weakly supervised learning has been already shown to be effective for improving zero-shot metrics in ImgSGG [3], indicating that a model trained on captions indeed increases its generalizability.
>
> **W2. Handling Long-term Relations**
> Here, we’d like to clarify the purpose of our proposed ADV module. This module is designed to detect relations that span across longer temporal windows. Specifically, the ADV module adaptively detects actions over various lengths by performing a fine-grained analysis of the vision-language score, as detailed in Section 2.3. For actions requiring short-term dependencies, the ADV module selects a few clusters to capture short-length actions. On the other hand, for actions requiring long-term dependencies, it selects several clusters to effectively capture longer actions. In other words, *our proposed module aims to capture relations spanning longer temporal windows*, addressing the concern regarding the handling of long-term relations raised by the reviewer.
>
> **W3. Occlusion and Tracking Limitations**
> We acknowledge that occlusions and visual artifacts can interfere with the formation of long tracklets. However, we’d like to clarify the training process in Section 2.4, i.e., Generation of Pseudo-Localized Scene Graphs. Specifically, when the unlocalized triplets parsed from segmented sentences are grounded to the corresponding frames aligned by the ADV module (refer to Section 2.3), some frames may fail to be grounded with those triplets due to occlusions or visual artifacts, resulting in incomplete tracklets. Nevertheless, *the frames successfully grounded with unlocalized triplets are still utilized for training the model, regardless of whether complete tracklets are formed*. Unlike video-level VidSGG, which relies on complete tracklets, the model for frame-level VidSGG is trained on a frame-by-frame basis. This approach makes tracking issues less critical compared to video-level VidSGG. As a result, concerns regarding the loss of tracklets are not a major issue in the weakly supervised video scene graph generation task.
>
>
>
>
>
> [1] Zero-shot Scene Graph Generation via Triplet Calibration and Reduction. Li et al. TOMM’23
> [2] Dynamic Scene Graph Generation via Temporal Prior Inference. Wang et al. MM’22.
> [3] LLM4SGG: Large Language Models for Weakly Supervised Scene Graph Generation. Kim et al. CVPR’24

---

> ### Author Response · Authors · 2024-11-22
> **(2/N) Author Rebuttal**
>
> **W4. High Computation Cost for Short Clips**
>
> |  Caption Dataset  | Avg. Video Length | W/C |       | N/C |       |  Mean |
> |:---------------:|:-----------------:|:---------------:|:-----:|:-------------:|:-----:|:-----:|
> |                 |                   |       R@20      |  R@50 |      R@20     |  R@50 |       |
> | Action Genome   |      29.9 sec     |      15.61      | 19.60 |     15.92     | 22.56 | 18.42 |
> | ActivityNet     |     117.3 sec     |      13.46      | 17.58 |     13.94     | 21.41 | 16.60 |
> (W/C: With Constraint, N/C: No Constraint)
>
> We agree that applying our method to very long video sequences may incur high computational costs due to repeated analysis of shorter clips split derived from the long video sequences. However, it is important to recall that our approach initially aims to train the VidSGG model using only video captions, even for short clips, rather than processing a wide range of video sequences to train the VidSGG model. Nevertheless, to address the reviewer's concern regarding the computational issue, we conducted an experiment on a longer video sequence dataset (i.e., ActivityNet), which has an average sequence length of 117.3 seconds - 4 times longer than that of the Action Genome dataset. Basically, to avoid high computation cost, we can opt to split very long video sequences into longer clips, leading to less repeated process. In this regard, the result in the table indicates that based on the comparison of video  lengths betwee the Action Genome and ActivityNet dataset, it is feasible to split the very long video sequences into 4 times longer clips, thereby reducing computation costs by a factor of 4, with a slight performance drop. Given that the ActivityNet caption is an external caption dataset, we think that this minor performance loss is considered acceptable.
>
> **W5. Missing Important References**
> We thank the reviewer for suggesting important references. Following the suggestion, we have included references to the video-level VidSGG work in the Related Work section (Line 492) and weakly supervised video grounding work in the Future Work section (Line 1172).
>
> **Q1. Experiment on Different Datasets**
> We acknowledge the importance of validating our method across different datasets to demonstrate its generalizability. To this end, we conducted an experiment on the VidHOI dataset, which is a frame-level VidSGG dataset, and please note that this has been originally included in Appendix C. In summary, for the VidHOI dataset, our proposed framework still outperformed the naive approach (i.e., WS-ImgSGG) discussed in Section 1. For ease of reference, this can be found on Line 327 in the main paper.
>
> Please note that as the VidOR dataset suggested by the reviewer is a *video-level VidSGG dataset*, it is not an appropriate dataset for evaluating the frame-level VidSGG models like ours. Hence, we instead conducted experiments with the VidHOI dataset, which is a frame-level VidSGG dataset.

---

> ### Author Response · Authors · 2024-11-22
> **(N/N) Author Rebuttal**
>
> **Q2. Handling Noisy or Ambiguous Captions**
> |        Method        | # Video | W/C |       | N/C |       |  Mean |
> |:--------------------:|:-------:|:---------------:|:-----:|:-------------:|:-----:|:-----:|
> |                      |         |       R@20      |  R@50 |      R@20     |  R@50 |       |
> | VSNLS                |  7,454  |      15.61      | 19.60 |     15.92     | 22.56 | 18.42 |
> | VSNLS - low quality  |  3,575  |      14.73      | 18.84 |     15.14     | 22.35 | 17.77 |
> | VSNLS - high quality |  3,575  |      15.21      | 19.24 |     15.58     | 22.55 | 18.15 |
> (W/C: With Constraint, N/C: No Constraint)
>
> To address the ambiguous caption causing the coreference issue, we provided the following prompt in the TCS module: *Note that you should specify the objects for the pronouns used in each of these sentences*. For the effectiveness of this prompt, we have discussed the effectiveness of this prompt in Appendix B. For ease of reference, this can be found in Line 204 of the main paper.
>
> Furthermore, we conducted additional experiments to evaluate ours method’s robustness against noisy captions.  For the noise criteria, we used the relevance score ranging from 1 to 7 given in the Charades datasets, from which the Action Genome dataset was made. This score indicates that the higher the score, the better the caption’s quality. Based on this score, we divided into two groups: those with scores from 1 to 6 (i.e., low-quality caption group), and those with a score of 7 (i.e., high-quality caption group). Since more than 50% of the video captions scored 7, we randomly selected videos from the high-quality caption to balance the number with the low-quality caption group. As shown in the table, the model trained on low-quality captions still shows competitive performance, validating the robustness of our framework against caption noise. It’s worthwhile to note that despite the numerous noisy captions in the entire video dataset, using the full dataset yields the best performance, indicating that the issue of noisy captions can be suppressed by mixing them with high-quality caption datasets.

---

> > ### Comment · Reviewer_aNnm · 2024-11-25
> > **Final Recommendation**
> >
> > Thanks for the detailed responses. The authors mentioned the differences between frame-level and video-level VidSGG, which should be emphases in the paper. And I have raised my previous scores. Hoping authors can release the well-organzied project code later and benefit the community.

---

> ### Author Response · Authors · 2024-11-25
>
> Thank you for your positive score! I'll make sure to emphasize the fame-level VidSGG in the introduction and will make the code available to the public.

---

### Official Review · Reviewer_QPb2 · 2024-11-10

**Soundness:** 3
**Presentation:** 3
**Contribution:** 3
**Rating:** 6
**Confidence:** 4

**Summary:**

This paper presents VSNLS, a weakly supervised approach for VidSGG that uses only video captions for supervision. The paper addresses two key challenges: temporality within captions and variability in action duration. The solution comprises three main components: Temporality-aware Caption Segmentation (TCS), Action Duration Variability-aware Caption-Frame Alignment (ADV), and a novel Pseudo-Labeling strategy based on Motion cues (PLM).

**Strengths:**

- Enabling VidSGG training using only video captions.
- Reduction in annotation costs compared to existing approaches.
- Well-identified key challenges in adapting image-based weak supervision to videos.
- Modular architecture with TCS and ADV components.
- PLM module handling negative action classes.
- Effective use of LLMs for temporal understanding.

**Weaknesses:**

- Limited discussion of LLM choice impact in TCS module.
- No detailed analysis of failure cases.
- An effective temporal consistency metric could be added.
- Analysis of performance on longer videos is missing.

**Questions:**

- Can you provide LLM selection criteria and impact on TCS performance?
- Can you clarify the robustness of K-means clustering for action duration estimation?
- Can you include an analysis of performance vs. video length?
- Can you provide an ablation experiment of TCS and PLM without ADV?
- Can you make use of a temporal consistency metric to make the analysis fair and comprehensive?
- What is the justification for the threshold value in the PLM? Have you done any experiments?

---

> ### Author Response · Authors · 2024-11-23
> **(1/N) Author Rebuttal**
>
> We sincerely thank the reviewer for the valuable and constructive feedback on our paper!
>
> **W1/Q1. Limited Discussion of LLM Choice impact in the TCS module**
> Regarding the discussion on the impact of LLM choice, we have already provided an analysis in Appendix D of the original manuscript, where ChatGPT is replaced with a smaller language model (i.e., Mistral 7B). In summary, we observed that the utilization of a larger language model performs better than that of a smaller language model, and both significantly outperform the naive approach (i.e., WS-ImgSGG). For the reader’s convenience, it is referenced in Lines 362-363 of the main paper. Please refer to Appendix D for a more comprehensive analysis.
>
> **W2. Detailed Analysis of Failure Cases**
> We discuss the failure cases of the TCS and ADV modules, respectively.
>
> For the TCS module, we found that it sometimes fails to segment the video captions in chronological order when the video caption is composed with ‘and’ to connect actions sequentially. For example, the phrase ‘A person is smoking a cigarette and looking at the phone’ can be confusing as it’s unclear whether the person is smoking while using the phone or finished smoking before looking at the phone. This issue may arise from relying on language-based captions without considering the visual features. We expect that this problem could be addressed in future work by using a Multimodal Large Language Model that analyzes visual features alongside segmenting the captions.
>
> For the ADV module, we found that it sometimes fails when the change between two actions is subtle. For example, in the scenario where a person smokes a cigarette before looking at the phone while the transition from ‘smoking’ to ‘looking at’ is marked by only slight facial or hand movement, the clustering algorithm within the ADV module may group both actions (i.e., smoking a cigarette and looking at the phone) into a single cluster since the visual features of the two actions are similar. However, what we aim to achieve is to group frames for each action into separate clusters. As a result, when aligning segmented sentences related to actions of ‘smoking a cigarette’ and ‘looking at the phone’ independently, the alignment may be ineffective. Inspired by the fact that a visual encoder trained in a self-supervised manner captures visual features in greater detail [1], we expect that using such a model could address this issue during the clustering process, thereby enabling the two actions to be clustered separately.
>
> [1] Eyes Wide Shut? Exploring the Visual Shortcomings of Multimodal LLMs. Tong et al. CVPR’24
>
> **W3/Q5. Introduction of Effective Temporal Consistency Metric**
> |                | Recall | Precision | F1 Score |
> |----------------|:------:|:---------:|:--------:|
> | WS-ImgSGG |  14.97 |   36.53   |   21.24  |
> | VSNLS          |  20.42 |   50.66   |   29.10  |
>
> We designed temporal consistency metrics (i.e., F1, Recall, Precision) to evaluate how accurately each action label extracted from the video caption aligns with the corresponding frame. To this end, we consider the ground-truth action labels within each frame as actual values, and action labels extracted from the video caption and assigned to the frames as predictive values. We then compute Recall, Precision, and F1 score. To directly measure the temporal consistency, we restrict the ground-truth action labels to only those found in each video caption. However, we report the metrics in the case where all ground-truth action labels are used in Section 4 of the revised manuscript.
>
> Using these metrics, we found that our proposed method significantly outperforms the naive approach discussed in Section 1, demonstrating the effectiveness of our proposed framework in terms of temporal consistency.
> Through the reviewer's suggestion regarding the temporal consistency metric, we could further clarify the effectiveness of our proposed method.

---

> ### Author Response · Authors · 2024-11-23
> **(2/N) Author Rebuttal**
>
> **W4/Q3. Analysis of Performance on Longer Videos**
> | Training Dataset |   Method  | Avg. Video Length | W/C |           | N/C |           |    Mean   |
> |:----------------:|:---------:|:-----------------:|:---------------:|:---------:|:-------------:|:---------:|:---------:|
> |                  |           |                   |       R@20      |    R@50   |      R@20     |    R@50   |           |
> |   Action Genome  | WS-ImgSGG |    29.9 seconds   |      10.01      |   12.83   |      9.02     |   14.05   |   11.48   |
> |                  | VSNLS     |                   |    **15.61**    | **19.60** |   **15.92**   | **22.56** | **18.42** |
> |       MSVD       | WS-ImgSGG |    9.5 seconds    |       6.22      |    8.03   |      7.69     |   12.31   |    8.56   |
> |                  | VSNLS     |                   |     **9.05**    | **11.31** |   **10.22**   | **16.60** | **11.80** |
> |    ActivityNet   | WS-ImgSGG |   117.3 seconds   |      10.86      |   14.47   |     10.07     |   15.80   |   12.80   |
> |                  | VSNLS     |                   |    **13.46**    | **17.58** |   **13.94**   | **21.41** | **16.60** |
> ###### *W/C: With Constraint, N/C: No Constraint
>
> To analyze the impact of video length, we conducted experiments using the ActivityNet caption dataset (average length: 117.3 seconds), which is approximately 4 times longer than the Action Genome caption dataset (average length: 29.9 seconds) and 12 times longer than the MSVD dataset (average length: 9.5 seconds). As shown in the table, we made the following two observations:
>
>   **1)** Regardless of video length, our proposed method consistently outperformed the naive approach (i.e., WS-ImgSGG). This indicates that our proposed method remains effective across videos of various lengths.
> **2)** When comparing the performance between the MSVD and ActivityNet datasets, aside from the benchmark dataset (i.e., Action Genome), we observed that both WS-ImgSGG and VSNLS achieved better performance on the longer ActivityNet dataset compared to the shorter MSVD dataset. We attribute it to the fact that longer videos allow the model to learn more diverse video content, thereby improving generalization, and provide more supervision as the duration of actions increases. In this context, despite the shorter video length of the Action Genome dataset compared to the ActivityNet dataset, our proposed method performs better on the Action Genome dataset. This is because the video distribution in the Action Genome dataset is more closely aligned with the test set, which is derived from Action Genome.
>
> We are deeply grateful to the reviewer for suggesting ways to conduct a more detailed analysis. To improve the quality of our paper, we have incorporated this result into Appendix N.
>
> **Q2. Robustness of K-Means clustering for action duration estimation**
> We argue that K-Means clustering shows robustness for action duration estimation. This is because adjacent frames showing similar actions would generally have similar visual representations, leading them to belong to the same cluster, while abrupt changes in action within frames would lay their visual features on different spaces, causing frames from that point onward to be allocated to different clusters (See Appendix H). In other words, when an abrupt change occurs over a short period, the small consecutive frames affected would be clustered, thereby capturing the short action duration by assigning the segmented sentence to the corresponding cluster. Similarly, this approach is applied to relatively long action duration. However, as discussed in Section 2.3, relying solely on K-Means clustering cannot effectively capture the longer action duration, leading us to propose the ADV module.
>
> Moreover, a recent work [2] employed K-means clustering for moment proposals in video moment retrieval tasks, supporting the robustness of K-means clustering for it.
>
> [2] Zero-Shot Video Moment Retrieval from Frozen Vision-Language Models. Luo et al. WACV’24.
>
> **Q4. Ablation study for TCS and PLM without ADV**
> |               | W/C |       | N/C |       |
> |:-------------:|:---------------:|:-----:|:-------------:|:-----:|
> |               |       R@20      |  R@50 |      R@20     |  R@50 |
> | VSNLS         |      15.61      | 19.60 |     15.92     | 22.56 |
> | VSNLS w/o ADV |      14.93      | 18.93 |     15.24     | 21.78 |
> ###### *W/C: With Constraint, N/C: No Constraint
>
> Following the suggestion, we conduct an ablation study to further clarify the impact of the ADV module. As shown in the table, we observe a decline in performance when the ADV module is excluded, demonstrating its effectiveness.

---

> ### Author Response · Authors · 2024-11-23
> **(N/N) Author Rebuttal**
>
> **Q6. Justification for the threshold value in the PLM and Experiment for it**
> Regarding the threshold value in the PLM, we have already conducted an experiment over various threshold values, detailed in Appendix J. In summary, we observed that as $\alpha$ increases, there is a gradual performance improvement, but it begins to fluctuate slightly beyond 5%. Nonetheless, we select $\alpha$ at 15% since it shows a slight improvement over 5%. For the readers’ convenience, we put the reference to line 365 of the main paper.

---

> > ### Comment · Reviewer_QPb2 · 2024-11-25
> >
> > Thank you for your detailed rebuttal. Please incorporate these critical elements from the rebuttal into the main paper instead of relegating them to the appendix:
> >
> > 1. TCS module: Add a brief but concrete example of a key failure case - specifically the ambiguity in segmenting captions with 'and' connectors.
> >
> > 2. Add the temporal consistency metric results as they provide crucial validation of the method's core capability.
> >
> > 3. ADV module: Briefly explain how the method handles varying action durations through the "steepest decline" criterion in similarity scores. This is critical to the method's effectiveness.
> >
> > I also have two brief follow-up questions for clarification:
> >
> > 1. For the clustering-based frame alignment in ADV, how sensitive is the "steepest decline" criterion for selecting relevant clusters to noise in the similarity scores? Did you consider or evaluate any alternative thresholding approaches?
> >
> > 2. In the temporal consistency metrics, there's a notable gap between Limited and Unlimited configurations. Could you briefly explain your interpretation of this gap and its implications for real-world applications?
> >
> > Looking forward to the changes and clarification on the follow-up questions.

---

> ### Author Response · Authors · 2024-11-27
>
> Thank you for your additional feedback! As per the reviewer’s suggestion, we have included three critical elements in the main paper as follows:
> Line 205-208: Failure case of the TCS module
> Line 469-481: Temporal consistency metric results
> Line 231-232: A Brief explanation of how the steepest decline handles varying action durations
>
> To clarify the reviewer’s understanding, we will address the two questions point-by-point.
>
> **Q1. Sensitivity of the “Steepest Decline” Criterion and Alternative Thresholding Approach**
> Regarding this question, we have added a new section titled “Discussion of the Steepest Decline in the ADV module” in Appendix Q.  Please refer to it for a detailed discussion.
>
> In summary, in Figure 11, we analyze the gap in the similarity score at the point of the steepest decline over the number of total clusters (i.e., $K$), which is proportional to the video lengths. Our analysis revealed that the gap at the point of the steepest decline converges to 0.2 as the number of total clusters increases. It demonstrates that even with noise in the similarity score, the steepest decline approach can reliably select relevant clusters using this relatively large 0.2 gap.
>
> In this regard, we raise a question: why not determine the relevant clusters through a point where the gap in the decline exceeds 0.2, instead of the steepest decline approach, since the threshold of 0.2 can reliably select relevant clusters? Therefore, in Figure 12, we conducted experiments where we set a fixed threshold of 0.2 across all videos to determine the relevant clusters. We observe that it shows competitive performance with our proposed steepest decline approach. However, as this fixed threshold increases, performance decreases since more irrelevant clusters are assigned, thereby introducing noise.
>
> Nevertheless, *we argue that to determine a suitable threshold, this fixed thresholding approach requires meticulous analysis of the gap at the point of steepest decline for each dataset of interest, whereas our proposed steepest decline can adaptively capture relevant clusters without the need to analyze each dataset.*
>
> **Q2. Explanation for Notable Gap Between Limited and Unlimited Configurations**
> The main reason for the performance gap between the Limited Configuration and the Unlimited Configuration is that the Ground-Truth (GT) video scene graph dataset contains far more numerous and detailed action labels than those found in video captions. Specifically, the GT scene graph dataset has action labels annotated between a person and various objects in every frame, and these are multi-labeled, which makes them more numerous than the action labels in the video captions. Thus, to improve performance in the Unlimited Configuration, consideration should extend beyond temporal consistency to include a broader range of action labels within video captions, reducing false negatives. However, we utilize existing video captions that focus on salient actions, excluding more detailed ones. That’s why recall in the Unlimited Configuration is lower than in the Limited Configuration.
>
> Consider a real-world application scenario where a person reads a book while sitting on a chair and then goes to the bathroom, but the human annotator or video captioning model simply describes the video as ‘a person reads a book and then goes to the bathroom.’ In fact, the video content would be more complex and diverse than this scenario suggests. The Limited Configuration only considers the alignment of two actions (i.e., reads a book and goes to the bathroom) with the corresponding frames, i.e., temporal consistency. Beyond this, the Unlimited Configuration requires an additional detail, such as ‘a person sits on a chair,’ which is not originally included in the video caption.
>
> In this regard, we focus only on temporal consistency, which is suitable for the Limited Configuration. However, we believe that future work could explore a new direction for generating captions that capture a broader range of actions in detail while capturing the temporal consistency, which is suitable for the Unlimited Configuration.

---

> > ### Comment · Reviewer_QPb2 · 2024-11-28
> >
> > Thank you for the changes and clarification. I would support the acceptance of the paper and maintain my initial score.

---

### Meta-Review · Area_Chair_LoMW · 2024-12-14

**Metareview:**

This paper proposed to a weakly supervised VidSGG with Natural Language Supervision (VSNLS) framework, which can reduce the high cost of manual annotation. Two modules are specially designed to address two key challenges related to video data, i.e. temporality within captions and variability in action duration. The challenges for weakly supervised VidSGG are well identified and addressed. The weakness is that the proposed method does not contribute to any core learning methods. All reviewers leaned to accept this paper, I recommend accept to it.

**Additional Comments On Reviewer Discussion:**

The concerns raised by reviewers include insufficient experiments analysis, unclear experiment details, method limitations, reference missing, etc. The authors’ rebuttals addressed these concerns well, and two reviewers raised their ratings. The other two reviewers maintained their positive ratings.

---

### Decision · Program_Chairs · 2025-01-22

Accept (Poster)